# Disordered clock protein interactions and charge blocks turn an hourglass into a persistent circadian oscillator

Meaghan S. Jankowski [1], Daniel Griffith [2], Divya G. Shastry[1], Jacqueline F. Pelham [1], Garrett M. Ginell [2], Joshua Thomas[1], Pankaj Karande [3,4], Alex S. Holehouse [2,5] & Jennifer M. Hurley [1,4] ✉

Organismal physiology is widely regulated by the molecular circadian clock, a feedback loop composed of protein complexes whose members are enriched in intrinsically disordered regions. These regions can mediate protein-protein interactions via SLiMs, but the contribution of these disordered regions to clock protein interactions had not been elucidated. To determine the functionality of these disordered regions, we applied a synthetic peptide microarray approach to the disordered clock protein FRQ in *Neurospora crassa*. We identified residues required for FRQ's interaction with its partner protein FRH, the mutation of which demonstrated FRH is necessary for persistent clock oscillations but not repression of transcriptional activity. Additionally, the microarray demonstrated an enrichment of FRH binding to FRQ peptides with a net positive charge. We found that positively charged residues occurred in significant "blocks" within the amino acid sequence of FRQ and that ablation of one of these blocks affected both core clock timing and physiological clock output. Finally, we found positive charge clusters were a commonly shared molecular feature in repressive circadian clock proteins. Overall, our study suggests a mechanistic purpose for positive charge blocks and yielded insights into repressive arm protein roles in clock function.

Circadian clocks have evolved as an adaptive mechanism to anticipate daily environmental changes and are recognized as an important regulator of the cellular environment amongst eukaryotes[1]. The presence of cell-intrinsic circadian molecular rhythms is conserved across species and broadly coordinates physiology and behavior, such that appropriate activities occur at biologically advantageous times. Because the circadian clock is so tightly intertwined with essential organismal systems, a disrupted circadian clock is detrimental to organismal fitness, as exemplified by increased disease rates in humans that have been exposed to chronic circadian disruption[2].

The underlying architecture of this cellular circadian clock in higher eukaryotes (including fungi, insects, and mammals) is here defined as an oscillator composed of the core molecular clock, a Transcription-Translation Feedback Loop (TTFL), made up of a positive arm transcriptional-activating protein complex and a negative arm repressing protein complex (Fig. 1a), and ancillary feedback loops[3]. Despite an understanding of clock regulation effected through transcriptional activation by positive arm proteins, questions remain regarding the fundamental biophysical mechanisms of negative arm feedback and circadian post-transcriptional regulation[4].

[1]Department of Biological Sciences, Rensselaer Polytechnic Institute, Troy, NY 12180, USA. [2]Department of Biochemistry and Molecular Biophysics, Washington University School of Medicine, St. Louis, MO 63110, USA. [3]Department of Chemical and Biological Engineering, Rensselaer Polytechnic Institute, Troy, NY 12180, USA. [4]Center for Biotechnology and Interdisciplinary Sciences, Rensselaer Polytechnic Institute, Troy, NY 12180, USA. [5]Center for Biomolecular Condensates, Washington University in St. Louis, St. Louis, MO 63110, USA. ✉e-mail: hurlej2@rpi.edu

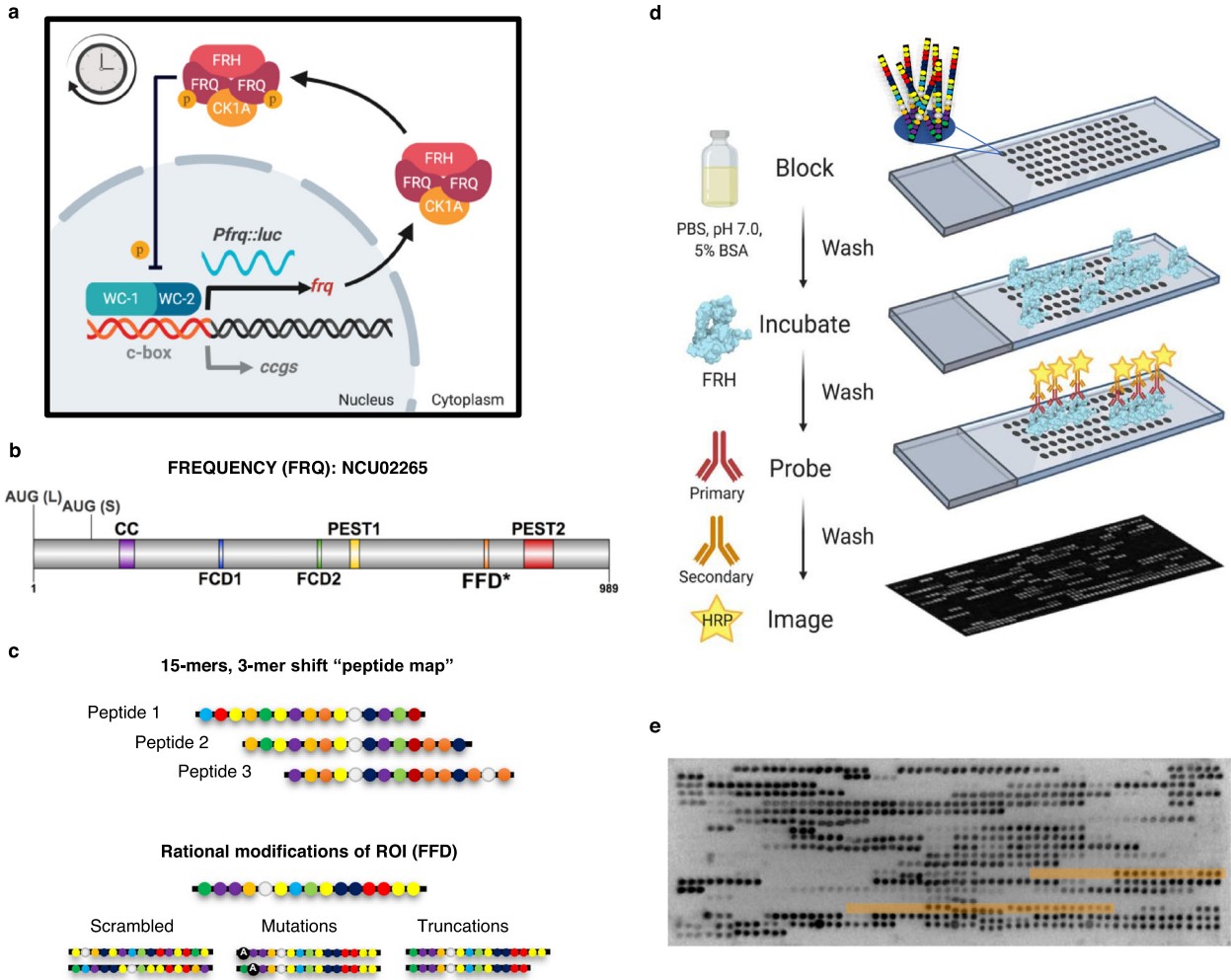

**Fig. 1 | LOCATE recapitulates a known FRQ SLiM. a** Schematic of the molecular clock in *N. crassa*. The transcriptional activators WC-1 and WC-2 activate *frq* transcription. Once translated, FRQ binds CK-1A and FRH leading to repression of WC-1/WC-2 activity at the *frq* promoter, presumably by phosphorylation, closing the feedback loop. The robust oscillation of the clock is represented by the continuous oscillation of luciferase expressed from the *frq* promoter in our reporter strain. FRQ (FREQUENCY), CK-1A (CASEIN KINASE 1a), FRH (FRQ-INTERACTING RNA HELICASE), *Pfrq frq* promoter, *luc luciferase*, c-box clock box, *ccgs* clock-controlled genes, p phosphorylation. **b** FRQ protein topology with known binding domains highlighted, AUG (L) Long FRQ start codon, AUG (S) Short FRQ start codon[103], CC Coiled-coil region[106], FCD-1/2 FRQ-CK-1A interaction Domain 1/2[34], PEST 1/2 Proline-Glutamic Acid-Serine-Threonine rich region 1/2[107], FFD FRQ-FRH interacting Domain, * region as defined by ref. 18. **c** The design of the FRQ LOCATE peptide library based on a 15-mer "peptide map" scan, shifting by 3 a.a., through the primary sequence of FRQ. Rational modifications were designed for Regions of Interest (ROI) using scrambled sequences, mutations, and truncations. **d** The experimental workflow for the LOCATE method; blocking, incubating with FRH, probing using an FRH-specific antibody, and chemiluminescent detection. **e** Example of the Library I FRQ microarray challenged with ~10 nM FRH and probed with anti-His antibodies, then image inverted for relative density analysis. Regions highlighted in orange are peptides including the FFD region (3-mer shift vs. 1-mer shift). The experiment was repeated a total of three times with similar results (see Supplementary Fig. 1d–e for correlation plots). Source data are provided at [https://doi.org/10.17632/7hgspb5gn7.1]. See also Supplementary Fig. 1. Image in (**b**) made with DOG (v2.0)[108], while images in (**a**) and (**d**) were created with BioRender.com.

A common feature of TTFL proteins, particularly those in the negative arm, is that the proteins are enriched for Intrinsically Disordered Regions (IDRs)[5]. IDRs are protein regions characterized by an inability to adopt a stable tertiary structure; instead, IDRs exist in a collection of interconverting conformations[6]. IDRs participate in diverse regulatory pathways, are frequently subjected to post-translational modifications (PTMs), and can contain short linear peptide motifs of about 5–15 amino acids, termed Short Linear Motifs (SLiMs), that permit interactions with domains in other proteins[7,8]. These features allow IDRs to interact with proteins across space and time, making them useful tools in timing the TTFL[5]. While genetic studies of negative arm clock protein functional regions have historically focused on conserved domains, contemporary work has shown that IDRs in clock proteins contribute to clock function. For example, IDRs within the C-terminus of mammalian Cryptochrome (CRY) and the C-terminal region of BMAL1 gate the timing of interactions with other core clock proteins[9,10] (see ref. 5 for a general review on the importance of disordered regions in clock proteins). Though we know that these IDRs play important roles in mediating clock functions, the disordered nature of clock proteins can hamper the global identification and molecular characterization of potential functional IDRs and the SLiMs that they may contain[11,12].

Our limited global understanding of how clock proteins mediate molecular recognition necessitates the identification of binding interfaces in an unbiased and high-throughput manner. Recently, synthetic peptide microarrays have been used to identify SLiMs within a single IDR that are bound by a more structured domain, or more generally to identify unknown interactors for proteins with IDRs, suggesting a protein microarray approach could be extended to investigate binding behavior across the IDRs of clock negative arm

proteins[13,14]. We hypothesized that given that negative arm proteins across different taxa are known to participate in dynamic macro-molecular protein complexes both within and beyond the clock, likely enabled by their IDRs, a microarray approach could identify IDR-based SLiMs involved in both the TTFL and post-transcriptional clock output to create a more complete picture of IDRs that are important for clock function[5,15,16].

Here, we report the use of the <u>L</u>inear m<u>o</u>tif dis<u>c</u>overy using r<u>at</u>ional d<u>e</u>sign (LOCATE) approach, which employed rationally designed printed synthetic peptides based on the sequence of the highly disordered negative arm clock protein FREQUENCY (FRQ) from the clock model organism *Neurospora crassa* (*N. crassa*)[17]. To validate our LOCATE method, we tested the FRQ-based microarray with its well-known stabilizing partner protein, FRQ-interacting RNA Helicase (FRH)[18,19]. LOCATE successfully identified a previously published SLiM that binds FRH, but also extended the SLiM to include additional essential residues[18,19]. This updated SLiM is essential for the interaction between FRQ and FRH, which was not necessary for repression as previous work suggested, but instead facilitated the conversion of an hourglass-like timer into a persistent circadian oscillator. Our LOCATE method also revealed that FRH binds to FRQ-based peptides that are positively charged and that these positively charged residues are sig-nificantly clustered within the FRQ sequence. Intriguingly, positive charge clusters are also found within the clock proteins of higher eukaryotes (e.g., the functional ortholog PER protein family). Impor-tantly, we show that these positive charge clusters are essential for normal clock function. Disruption of just one positive charge "block" within FRQ significantly lengthened the core clock period and affected clock output but did not break the TTFL. In total, LOCATE-based identification of FRH-binding sites within FRQ provided insight into the FRH/FRQ interaction, affording a testable and mechanistic model for clock protein function in tuning circadian timing.

## Results

### LOCATE recapitulates a known FRQ SLiM

We sought to better understand the sequence-determinants of IDR-mediated molecular recognition between clock proteins using the 989-residue, largely disordered, protein FRQ. FRQ is a central player in the negative arm of the *N. crassa* circadian clock and binds a collection of partners via many putative SLiMs (Supplementary Figs. 1a and 2d)[15,20]. FRQ's large size means working with full-length FRQ in vitro is technically challenging, making it difficult to assess how distinct regions contribute to molecular interactions. To circumvent this challenge, we constructed a 15-mer "peptide map" of sequentially overlapping peptides that scanned through the primary sequence of FRQ (NCU02265; Fig. 1b), shifting by 3 a.a. (Fig. 1c) (see "Methods" section)[17]. Our goal was to use this peptide map to deconvolve which regions of FRQ enable molecular interactions. Regions of interest were investigated by rationally designed peptides (e.g., scrambles, single-point mutations, and truncations)[17]. To validate the LOCATE approach, we examined relative binding intensities between the library and the FRQ Nanny protein, FRH (NCU03363), using *E. coli* expressed FRH (a.a. 100-1106, with a 6× His-tag) (Fig. 1d and Supplementary Fig. 1b, c)[19,21]. As a positive control, the LOCATE approach recognized the previous genetically identified FRQ-FRH domain (FFD, a.a. 774–782) (Fig. 1e, areas highlighted in orange), showing LOCATE is able to identify regions of interaction between FRQ and FRH[18].

### FRH interacts predominantly with positively charged FRQ pep-tides in the LOCATE assay

Beyond the identification of the previously known FFD binding domain, there were many other instances of interaction between FRH and the FRQ peptides, as shown by highly correlated results from technical replicates using different estimated FRH concentra-tions, as well as both anti-His and anti-FRH antibodies (Fig. 1e, and

Supplementary Fig. 1d, e). To further characterize the overall binding behavior between FRQ peptides and FRH in our LOCATE assay, we investigated the characteristics of the top 10% of FRH-binding FRQ peptides. These peptides were enriched in basic residues, revealing that FRH binding to FRQ peptides was skewed towards more positively charged peptides (Fig. 2a–c, Supplementary Fig. 1f). Overall, FRH binding correlated strongly with positive Net Charge Per Residue (NCPR) peptides, denoted as blue dots within the binding intensity graph (Fig. 3a). This corresponded with a previously solved FRH crystal structure, PDB5E02, that shows the surface electrostatic potential of FRH is mainly negative (Fig. 2d)[21]. Congruent with this, the relative position (or clustering) of oppositely charged residues, known as charge patterning, is an important sequence feature for some dis-ordered proteins[22–25]. We surveyed the linear distribution of NCPR across FRQ using a 10 a.a. scanning window to determine if charged residues were also highly clustered in FRQ. We found that positive and negative NCPR scores appeared to be grouped on FRQ, forming alternating positively and negatively charged seg-ments (Fig. 3a).

### Clusters of positive residues are commonly occurring features in negative arm clock proteins

Notably, previous work has highlighted how clusters of positively charged residues can play key roles in molecular recognition across a variety of systems[22,23,26–29]. Given the grouping of positive residues we saw within the FRQ sequence, we quantified the clustering of positively charged residues within FRQ to see if these were significantly clus-tered, as had been seen in other proteins with large IDRs[22–25]. To do so, we used the Inverse-Weighted Distance (IWD), a metric that quantifies the clustering of residues (Eq. 2 in "Methods" section)[30]. Comparing the IWD of positively charged residues within FRQ to the IWD calcu-lated from 10,000 randomly shuffled FRQ sequences (the null dis-tribution), we found that both positively charged residues and negatively charged residues, but not aromatic residues, in FRQ were significantly clustered (Fig. 3b and Supplementary Fig. 2a, b)[30–32].

To assess if the clustering of charged residues was conserved in FRQ across species, we investigated the sequence characteristics of 86 FRQ orthologs. Since orthologs differ in overall protein length and total number of residues of interest, the IWD calculation and null distributions were carried out separately for each FRQ ortholog and reported as a Z-score (see "Methods" section) to allow for direct comparison[31,32]. Most FRQ orthologs demonstrated significant positive charge clustering and to a lesser extent, negative charge clustering (Supplementary Fig. 2c). We next investigated charge clustering within the functional orthologs of FRQ in *D. melanogaster*, *M. musculus*, and *H. sapiens*, the PERIOD (PER) proteins. We found that all these PERs possess highly clustered positively charged residues (Fig. 3c), sug-gesting that positive charge clustering may be a consistent molecular feature amongst disordered negative arm clock proteins in higher eukaryotes[33]. To rule out that these positive charge clusters are simply common in the *N. crassa* proteome, we compared FRQ to an annotated set of *N. crassa* proteins (n = 1295, including cytoplasmic and nuclear proteins) and found that FRQ ranked in the top 7% for positive charge clustering (Supplementary Fig. 2d). This data showed that while not exclusive to clock proteins, significant positive charge clustering is both unusual and a shared feature of clock negative arm proteins, consistent with a role for positive charge clusters in clock function.

To investigate any potential role for positive charge clusters in clock function, we next identified which specific regions in FRQ have significant groupings of positively charged residues. To do so, we used a method that identifies charge "blockiness", which reports the num-ber of charge blocks in an IDR (defined as 10 a.a. subregions with an NCPR of ≥0.5 or ≤−0.5)[22]. Both charge blockiness and charge IWD are distinct from charge patterning parameters such as kappa (κ) or Sequence Charge Decoration (SCD), which quantify how evenly

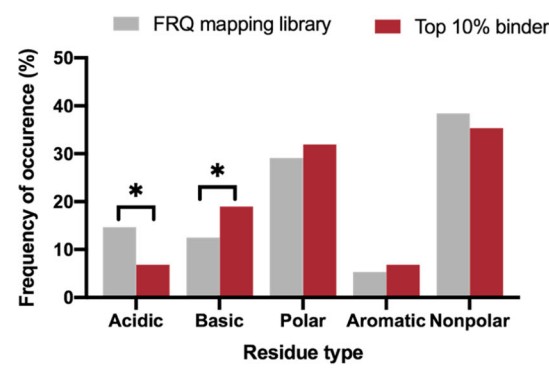

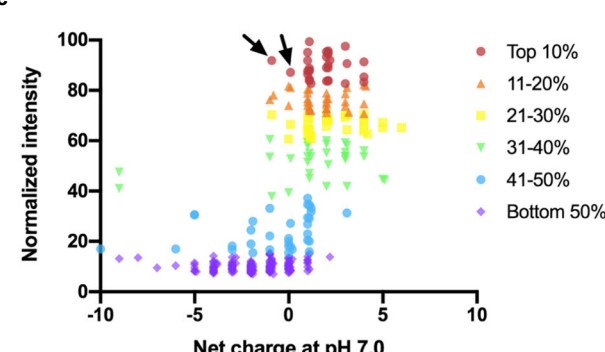

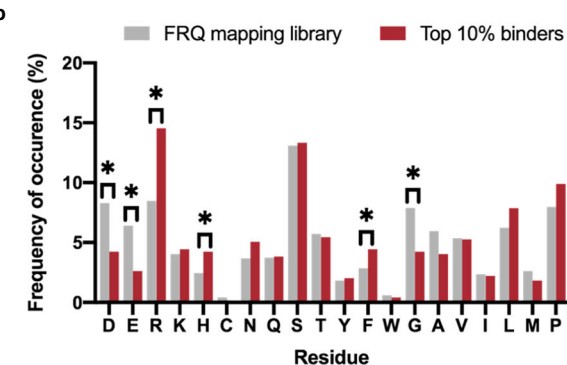

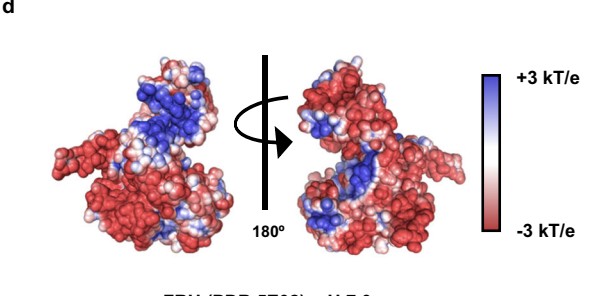

**Fig. 2 | The LOCATE method uncovers that FRH predominately binds to positively charged FRQ peptides. a** Comparison of the amino acid types in the overall FRQ mapping library and the amino acid types in the top 10% of FRH-binding peptides. Acidic = D and E; basic = R and K; polar = H, C, N, Q, and S; aromatic = Y, F, and W; and nonpolar = G, A, V, I, L, M, and P. **b** Comparison of the amino acids in the overall FRQ mapping library and the amino acids in the top 10% of binding peptides. The statistical significance of frequency changes of residues C, Y, W, and M was not determined due to low residue occurrence in the sample population. **a**, **b** were tested using a two-tailed z-test for population proportions; * = p < 0.05. Exact test statistics and p-values are reported in the Source Data file. **c** Ranking of FRQ

mapping peptides by normalized binding intensity and estimated net charge at pH 7.0. Peptides are colored by their binding decile as indicated. Arrows denote two peptides containing the FFD with an overall neutral charge. See Supplementary Fig. 1f for a densitometric view of this data. Note that (**a**), (**b**), and (**c**) are based on average normalized intensities of 3-mer mapping peptides from Library I using the anti-His and anti-FRH experiments both using ~10 nM FRH. Source data are provided as a Source Data file. **d** The calculated surface electrostatic potential map for FRH (PDB5E02; see "Methods" section). Blue denotes positive charge while red denotes negative charge, ranging from +3 kT/e to −3 kT/e. FRQ (FREQUENCY), FRH (FRQ-INTERACTING RNA HELICASE).

distributed oppositely charged residues are across an entire IDR[24,25]. When we applied this block definition to FRQ, the analysis highlighted several discrete blocks of positive (red lines above NCPR plot) or negative charge (blue lines below NCPR plot) (Fig. 3a), consistent with our NCPR analysis and confirming the presence of clusters/blocks of positive and negative residues. Interestingly, the phosphosites that occur on FRQ over the circadian day have a greater potential to alter negative charge blocks rather than positive charge blocks along the length of FRQ, and overlapped with many predicted downstream SLiMs (Supplementary Fig. 2e, f)[15,20].

**Blocks of positively charged residues affect circadian timing and output but are not essential for in vivo FRQ/FRH interaction**
The repeated presence of positive charge patterning in our in vitro and bioinformatics work implicated that these blocks of positive charge could be important for FRQ function in vivo. To directly test this, we mutated residues (*KKK, a.a. 315–317) to ablate a positive charge block that was bound by FRH, but also adjacent to the genetically identified CK-1A interaction site known as FCD-1 (a.a. 319–326) (Fig. 3a and Supplementary Fig. 3a)[34]. We targeted VHF-tagged (V5, 10-His, 3-Flag) alleles of Wild-type (FRQ^VHF) and KKK315AAA (FRQ^KKK/AAA) substitutions to the cyclosporin (csr-1) locus of an frq KO strain with a banding and frq promoter luciferase-reporter background[35–37]. Banding mutants (ras-1^bd) lay a dense band of visible spores (a conidial "band") once per clock cycle. By growing Neurospora in a long glass tube known as a race tube, dense bands of fungal growth (conidial bands) appear with

circadian periodicity along the race tube when the clock is running. This, in turn, allows for the direct analysis of the FRQ^KKK/AAA mutation on overt clock rhythms[38]. Conidial band formation demonstrated that, while the FRQ^VHF strain had a functional clock, the FRQ^KKK/AAA strain had no overt clock rhythms (Fig. 3d and Supplementary Fig. 3b).

Banding is a measurement of clock control of physiology but is not always a direct representative measurement of the activity of the core TTFL[39]. Therefore, we next assayed frq promoter activity using the luciferase reporter present in the strains as a proxy for the TTFL, as the activation of the frq promoter in this strain also leads to the transcription/translation of luciferase[40]. Surprisingly the FRQ^KKK/AAA strain displayed a longer, non-circadian oscillation (period = 34.88–37.87 h, n = 3 transformants and n = 6 wells each) compared to the FRQ^VHF strain, which showed a 21.8 h oscillation in frq promoter activity according to an analysis with the oscillation detection software ECHO (Fig. 3e and Supplementary Fig. 3c)[41]. This suggests the clock in the FRQ^KKK/AAA strain is still oscillating but has been decoupled from the physiological regulation that results in banding.

To determine if the lengthening of the core clock period and the uncoupling of the clock to physiological output was due to changes in FRQ binding to FRH or CK-1A, we carried out co-immunoprecipitation on FRQ using FRQ^VHF and FRQ^KKK/AAA strains where CK-1A had been fused to an HA epitope tag (Supplementary Fig. 3d). We found that FRQ^KKK/AAA binding to FRH was not significantly different when compared to FRQ^VHF binding of FRH. When we looked at CK-1A interaction using anti-HA, overall levels of CK-1A binding in FRQ^KKK/AAA were also

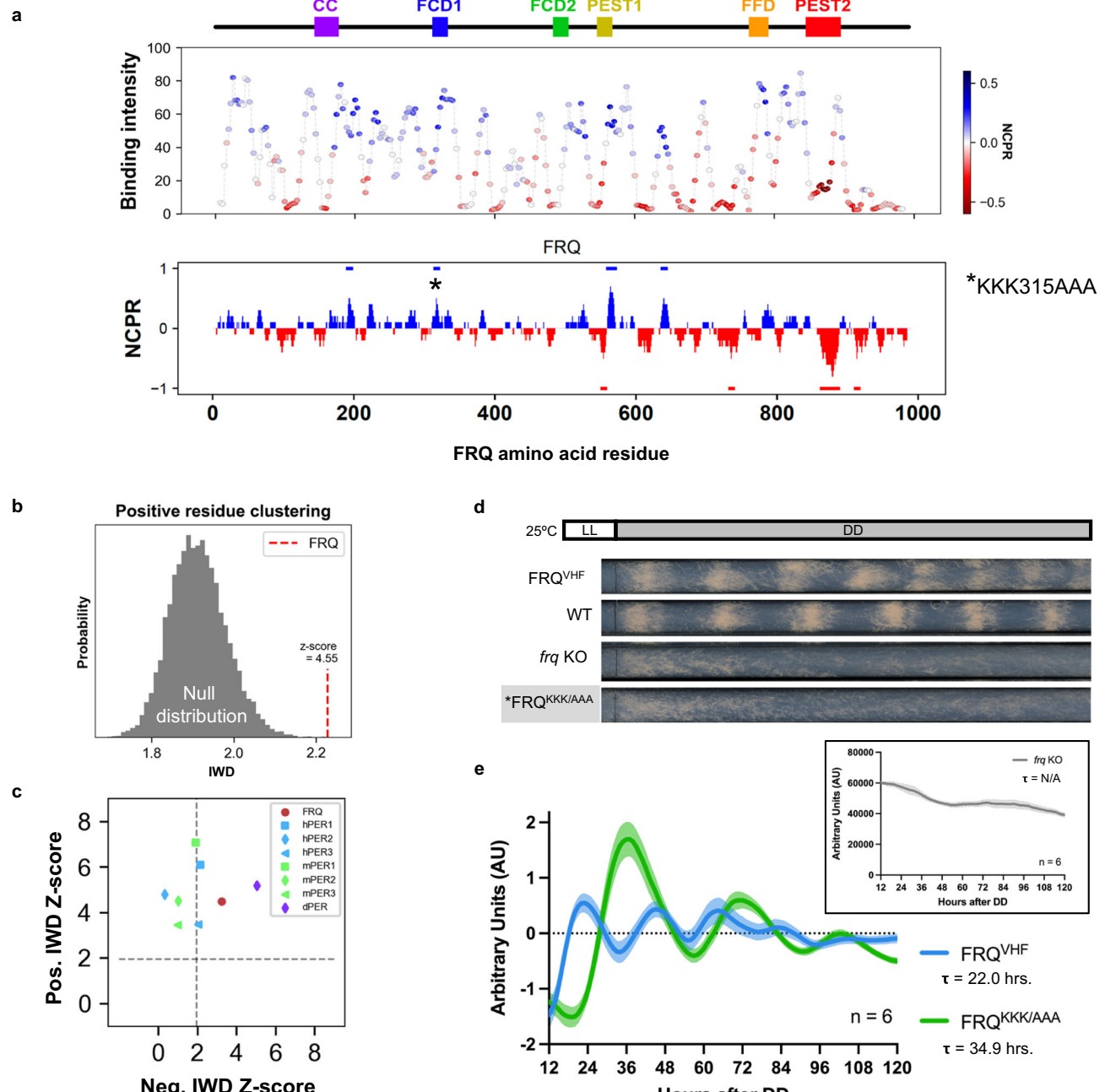

**Fig. 3 | There are significant charge clusters within FRQ and mutating a positive charge block affected clock period and output. a** Normalized average FRH-binding intensity (-10 nM FRH, anti-His), with peptides color-coded by Net Charge Per Residue (NCPR) as shown in the legend, and local NCPR (using a 10 residue sliding window) plotted against the sequence and known domains of FRQ (abbreviations as in Fig. 1). Blue lines above the NCPR plot denote positive charge blocks, while red lines below denote negative charge blocks (see "Methods" section). The location of the KKK315AAA mutation is shown using an asterisk (*). See also Supplementary Fig. 2e for a comparison of NCPR when FRQ is hyper-phosphorylated. **b** Positive residue clustering in FRQ (red) compared to a null distribution (gray) of 10,000 random FRQ sequence shuffles. Residue clustering is computed using the Inverse-Weighted Distance (IWD) metric (see "Methods" section). A normalized Z-score was computed by comparing WT FRQ's IWD to the mean and standard deviation of the null distribution. As FRQ's Z-score is >1.96, it has significant positive clusters at $p < 0.05$ based on the null distribution. See Supplementary Fig. 2a, b for this same analysis of negative residue and aromatic residue clustering. **c** The normalized Z-score for the calculated IWD parameter for positive and negative residues for FRQ (red circle) and functional PER orthologs (see legend). The Z-score was computed individually for each ortholog, using separate, sequence-specific null distributions. The dashed lines denote Z-score >1.96 (two-tailed $p < 0.05$). See Supplementary Fig. 2c, d for a comparison of FRQ orthologs and a comparison of FRQ to other proteins in *N. crassa*. **d** Representative race tubes of FRQ^VHF, Wild-type FRQ (WT), *frq* KO, and FRQ^KKK/AAA strains grown in constant dark (DD). See Supplementary Fig. 3b for $n = 3$ biological replicates of FRQ^KKK/AAA from one experimental round. **e** Smoothed, linear detrended, and normalized average and standard deviation values of $n = 6$ wells from FRQ^VHF (tagged wild-type) vs. FRQ^KKK/AAA strains with a luciferase reporter of *frq* promoter activity (*frq*c-box-*luc*+) background grown in DD in a 96-well format in the presence of luciferin. Inset shows smoothed data from a *frq* KO strain, for comparison. See also Supplementary Fig. 3c for other biological replicates. FRQ (FREQUENCY), FRH (FRQ-INTERACTING RNA HELICASE), τ period as fitted by ECHO v3.22. Source data are provided as a Source Data file, at https://zenodo.org/records/10793684 and https://doi.org/10.17632/7hgspb5gn7.1.

not significantly changed compared to wild-type FRQ$^{VHF}$ levels of CK-1A interaction (Supplementary Fig. 3d, e). Furthermore, there appeared to be no change in the level of phosphorylation of FRQ, suggesting there was no significant change in CK-1A kinase activity on FRQ (Supplementary Fig. 3d).

It was possible that these mutations modified the conformational ensemble of FRQ to affect function rather than the effect being based on charge. To establish if this was a likely scenario, we carried out all-atom simulations of a 50 a.a. region of FRQ centered on the FCD-1 region. In comparison to the wild-type peptide, the KKK315AAA mutation had a minimal effect on solvent-accessible surface area (SASA) of the FCD-1 region (Supplementary Fig. 4a). While we did observe some changes in long-range interactions, these were driven by the loss of repulsion between 315-KKK-318 and other positively charged residues elsewhere in the peptide (Supplementary Fig. 4b, c). The importance of overall charge over specific ensemble features was further supported by the consistency of binding of FRH in the LOCATE scrambles of this region (Supplementary Fig. 4f). Taken together, our simulations suggest that the loss of these three lysines does not significantly alter the accessibility of the nearby FCD-1 motif and that the ablation of the charge in this region underlies the noted clock effects.

## Essential residues beyond the canonical FRH-binding SLiM are needed for FRQ/FRH binding

Given the lack of disruption to FRQ/FRH binding upon the mutation of a positive charge block, we investigated the LOCATE results to determine if there were peptides that bound FRH but were not positively charged. While most FRH interactions with FRQ peptides in the LOCATE assay were correlated with regions of positive charge, consistent with a role for electrostatics in these interactions, a few FRQ peptides showed relatively high binding to FRH, but had a net neutral charge (see arrows in Fig. 2c). These peptides mapped to the FRQ/FRH interaction domain (FFD), a region on FRQ that was previously identified as the necessary interaction region with FRH[18]. This canonical FFD binding motif, or SLiM, occurs within FRQ a.a. 774–782 and is composed of multiple hydrophobic/small residues flanked by charged residues (Fig. 4a)[18]. While the FFD SLiM is not strictly conserved among FRQ homologs, some positions toward the beginning are well-conserved, while at other positions similar physicochemical residues are retained, suggesting an evolutionary chemical signature (Fig. 4a and Supplementary Fig. 4d). When we analyzed the binding of FRH to scrambled FFD peptides that maintained the same amino acid composition and neutral net charge, we found scrambles with diminished binding (Supplementary Fig. 4e). A comparable analysis of the interaction near the FCD-1 region (described above, peptide MTDKEKKKLVVRRLE, predicted net charge +3) showed minimal changes in binding intensity upon shuffling (Supplementary Fig. 4f). Therefore, the FFD SLiM is distinct from other FRH-binding peptides in that it encodes some degree of sequence-specific binding.

SLiM-based protein interactions are often mediated by specific residues that play an essential role in binding, which are more defined or constrained (i.e. substitutions have a negative effect on binding) than other residues within the SLiM[42–45]. In our LOCATE analysis, we noted a sharp increase in binding when two arginine residues C-terminal of the canonical FFD motif entered the peptide window (a.a. 783–784, Fig. 4b). These double arginine residues, or similarly-charged residues (KR), were well-conserved in FRQ homologs (Fig. 4a)[46]. Truncations of peptides based on this region of FRQ showed these arginines were essential for FRH binding to the peptide microarray (Fig. 4c). Our LOCATE amino acid substitution matrix largely showed decreased FRH binding when these positions were substituted, especially the first arginine (Fig. 4d). While some substitutions of the

second arginine to other types of residues led to increased binding on the peptide microarray (Fig. 4d), the scanning data (Fig. 4b), the truncations (Fig. 4c), and the high degree of conservation of these residues (Fig. 4a) all suggested that the double arginines were a vital component of the FFD SLiM[47,48].

To verify the importance of these essential RR residues in vivo, we again targeted VHF-tagged (V5, 10-His, 3-Flag) alleles of FRQ to the csr-1 locus of an frq KO strain (Fig. 5a)[35,36]. We recreated a previously published alanine substitution for part of the FFD (FRQ$^{FFD2}$, VMLVTT777AAAAAA), and made additional double arginine to double alanine (RR783AA, FRQ$^{RR/AA}$) and double arginine to double histidine (RR783HH, FRQ$^{RR/HH}$) mutant strains (Fig. 5a)[18]. We tested the ability of these FRQ isoforms to interact with FRH in vivo using co-immunoprecipitation (Fig. 5b and Supplementary Fig. 5a). While FRQ$^{VHF}$ pulled down ample FRH, as did FRQ$^{RR/HH}$, as predicted by LOCATE and previous publications, both FRQ$^{FFD2}$ and FRQ$^{RR/AA}$ were unable to co-immunoprecipitate FRH (Fig. 5b and Supplementary Fig. 5a)[18].

To gain mechanistic insight into how the double arginine residues affected the binding of FRQ to FRH, and how the alanine mutations might have affected the solvent-accessibility of the FFD motif, we performed all-atom simulations of a 50 a.a. section of FRQ that centered on the FFD region[18] (Supplementary Fig. 5b–e, Supplementary Movie 1), including simulations of wild type (WT), RR783AA, RR783HH with neutral histidines, and RR783HH with positively charged histidines. RR783AA and RR783HH variants had a limited impact on the structural ensemble of the region outside of the specific position where the mutations occur (Supplementary Fig. 5b–d). Given the limited predicted impact of the RR783AA mutation on the local conformation (e.g., solvent-accessibility or SASA), our data suggests that the arginine residues work in combination with the hydrophobic residues to facilitate FRH binding to the FFD SLiM[49,50]. This implied that the primary binding site of FRQ and FRH at the FFD includes these essential RR residues.

## Binding to FRH is essential for persistent clock oscillations rather than repression in vivo

The functional consequences of deleting the residues in the previously published FFD are well-established, with a reduction in FRQ/FRH interaction correlated with a loss of FRQ stability[18,19,51]. This phenomenon has been attributed to FRH's function as a nanny protein, preventing the constitutive degradation of FRQ[19,52,53]. How FRH keeps FRQ from undergoing "degradation-by-default" is unknown but given that the FRQ$^{RR/AA}$ mutation led to a loss of FRH binding, we anticipated that FRQ$^{RR/AA}$ would phenocopy the FFD deletion mutant (FRQ$^{FFD2}$) in terms of FRQ stability.

To test how the essential arginine residues influenced FRQ degradation, we analyzed FRQ stability over time using cycloheximide (CHX)[19,52]. A two-way ANOVA showed significant differences in FRQ levels between strains and over time (Strain factor $F = 34.32$, $p < 0.0001$; Time factor $F = 79.75$, $p < 0.0001$). Specifically through Tukey's multiple comparisons test, the FFD alanine replacement mutant FRQ$^{FFD2}$ showed a significant decrease in FRQ stability compared to FRQ$^{VHF}$, as expected ($p < 0.0001$; Fig. 5c and Supplementary Fig. 6a)[18,19]. However, to our surprise, FRQ$^{RR/AA}$ was significantly more stable than FRQ$^{FFD2}$ ($p < 0.0001$), despite the loss of FRH binding (Fig. 5b, c and Supplementary Fig. 6a). These data implied that we had stabilized FRQ by mutating the double arginines within the extended FFD SLiM that is usually shielded by FRH.

In addition, we noted in the FRQ$^{RR/AA}$ strain that the overall levels of the positive transcriptional activator White Collar-1 (WC-1, NCU02356) (Fig. 5d and Supplementary Fig. 6b), were unusually high compared to the low WC-1 levels typically seen in strains where FRQ cannot close the feedback loop and stabilize the White Collar Complex (WCC) via phosphorylation (FRQ$^{RR/AA}$ vs. frq KO Tukey's adjusted

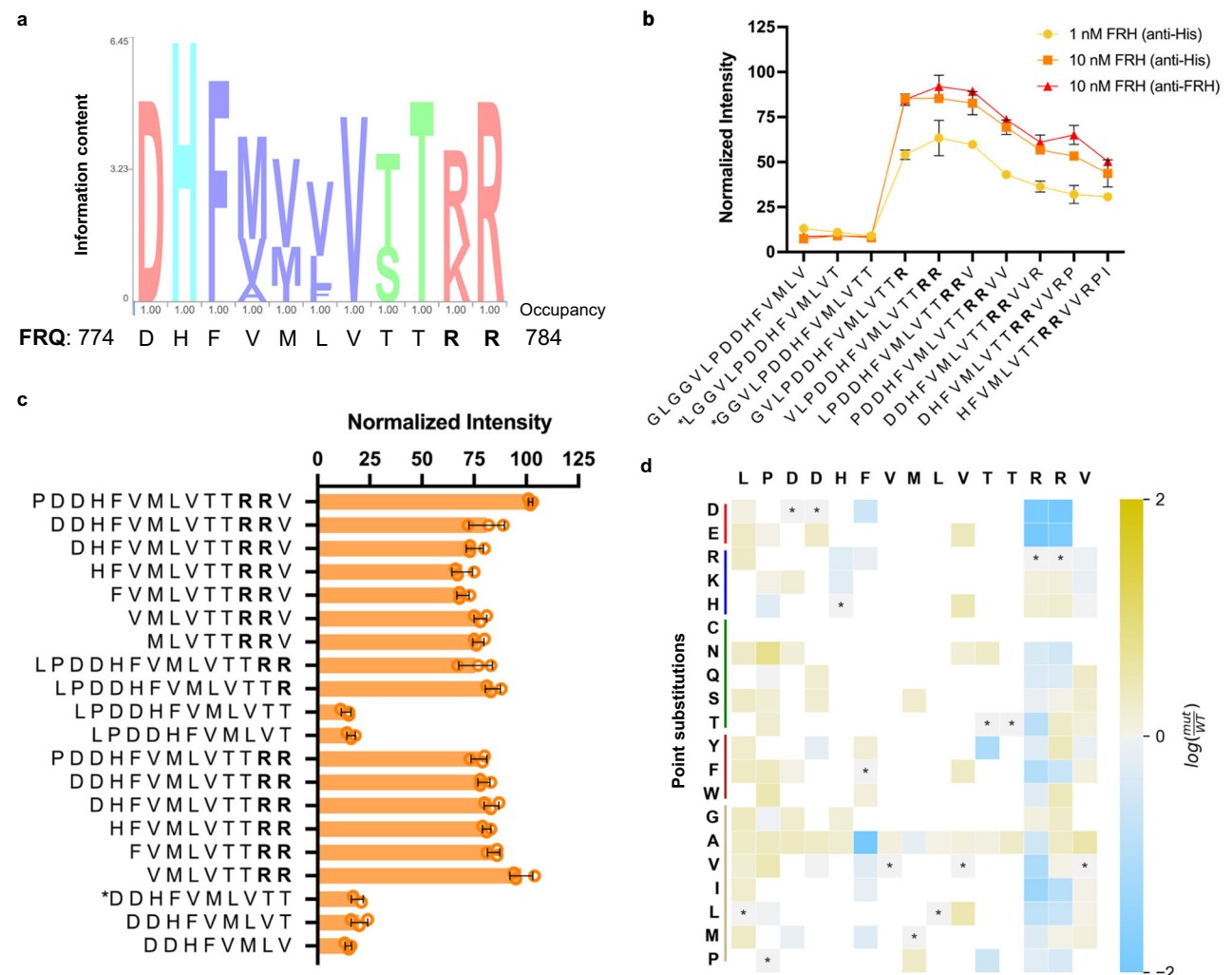

**Fig. 4 | LOCATE-identified residues are essential for FRQ/FRH interaction, extending the FRH SLiM. a** Weighted observed residues at each position for the FFD region of FRQ over 10 FRQ homologs (species shown in Supplementary Fig. 4d), with *N. crassa* FRQ residues listed below. Color denotes residue type (pink is charged, blue is histidine or tyrosine, purple is small/hydrophobic, and green is a hydroxyl/amine following the ClustalX scheme). Occupancy refers to the percentage of the position that was filled. **b** The average (SD) normalized intensity of FRH binding to the stated FRQ-based peptides from Library I, from three different experiments using either different FRH concentrations or different antibodies to visualize results. **c** Average (SD) normalized microarray intensity of FRH binding to different FFD-based peptide truncations. Also, see Supplementary Fig. 4e for results of FFD-based scrambles. **d** Point substitution analysis of the FFD SLiM with

the original peptide listed along the top and the tested amino acid point substitution along the left-hand side. Grayed boxes with stars denote the wild-type residue at that position, yellow denotes an increase in FRH associated with a given substitution, and blue denotes a decrease in FRH-binding intensity. Note that the scale is log₂ of mutant peptide (mut)/wild-type peptide (WT). White boxes correspond to amino acids substitutions that were not tested. Unless otherwise noted, results in this figure are based on Library II peptides, incubated with ~100 nM FRH and visualized with anti-His. $n = 3$ technical replicate spots except for peptides denoted with * where $n = 2$ spots. In (b) $n = 2$ only for the * peptides in the 10 nM FRH (anti-FRH) experiment. FRQ (FREQUENCY), FRH (FRQ-INTERACTING RNA HELI-CASE). Source data are provided as a Source Data file and at [https://doi.org/10.17632/7hgspb5gn7.1].

$p$-value = 0.0027 and FRQ^RR/AA vs. FRQ^FFD2 Tukey's adjusted $p$-value = 0.0011[54]. Therefore, the high WC-1 levels suggested stabilized WC-1, meaning that FRQ^RR/AA could close the negative feedback loop without binding FRH.

To validate the ability of FRQ^RR/AA to close the feedback loop independently of FRH, we followed the banding phenotype of the above strains using a race tube assay. We found the FRQ^VHF and FRQ^RR/HH strains maintained a typical clock period (21.25 and 20.86 h respectively), whereas the FRQ^FFD2 and FRQ^RR/AA strains were arrhythmic (Fig. 6a and Supplementary Fig. 7a)[18]. However, when we tracked clock functionality at the molecular level using luciferase reporter strains as described above, we found that while the FRQ^VHF, FRQ^RR/HH, and FRQ^FFD2 strains mirrored their overt phenotypes, the FRQ^RR/AA strain showed a re-activation of *frq* promoter activity within the first 24 h that was then repressed and highly dampened on the

second day (FRQ^RR/AA Amplitude Change Coefficient (ACC) = 0.24 compared to FRQ^VHF ACC = 0.07 as estimated by ECHO, where $-0.15 \leq ACC \leq 0.15$ is here considered a persistent oscillation[41]; Fig. 6b and Supplementary Fig. 6c)[35,40]. This is consistent with FRQ^RR/AA closing the feedback loop but the TTFL not robustly reactivating on the second day.

To corroborate this finding, we assessed conidiation in race tubes in a 12 h light:12 h dark (12L:12D) entrainment regime, which will only band if FRQ is able to close the feedback loop. While the FRQ^VHF, *frq* KO, FRQ^FFD2, and FRQ^RR/HH strain 12L:12D results mirrored their overt clock phenotypes seen in constant dark, the FRQ^RR/AA strain regained banding rhythms in the 12L:12D regime (Fig. 6c and Supplementary Fig. 7b). This data is consistent with FRQ^RR/AA being able to repress the WCC without FRH, and that this hourglass circuit can be restarted each day by receiving a new round of light input (Fig. 6d).

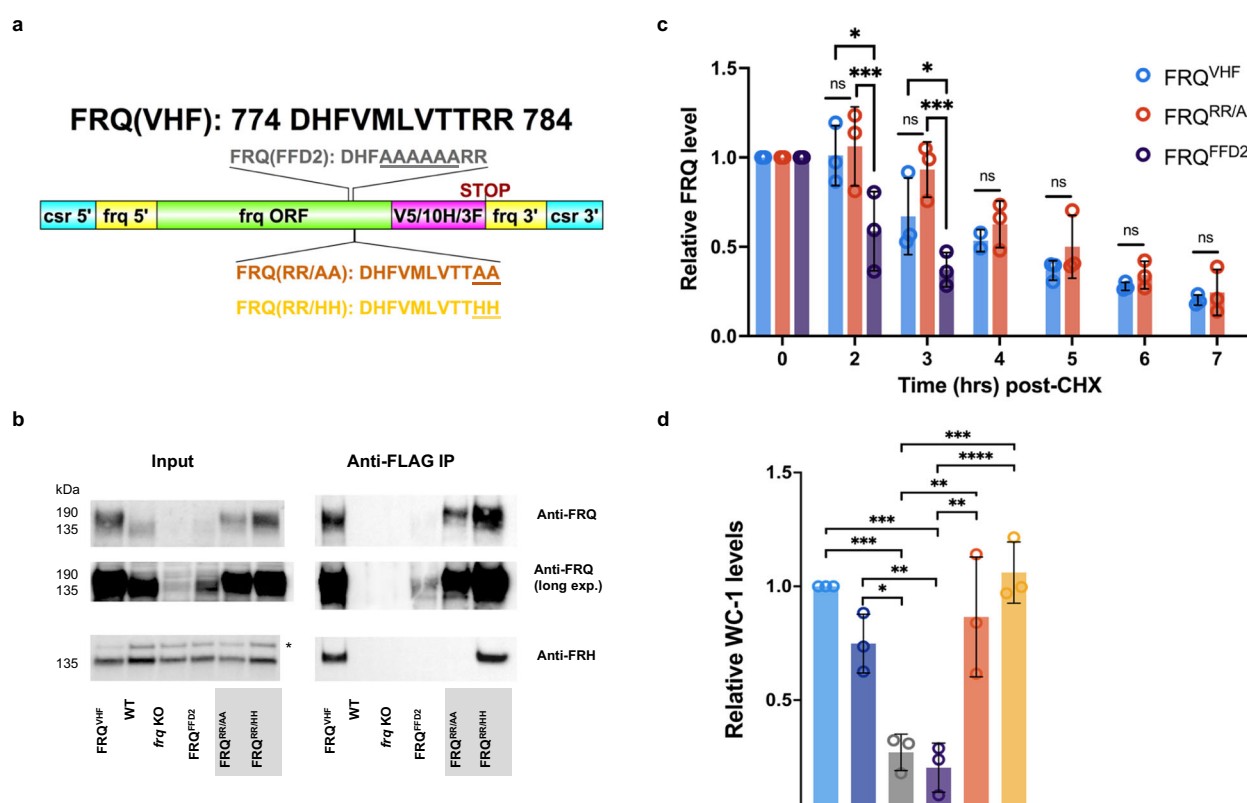

**Fig. 5 | In vivo mutation of the LOCATE-identified essential SLiM residues leads to a loss of FRQ-FRH interaction without a loss of repression. a** Schematic of the genetic mutations made within the FRQ-FFD region, plotted using DOG (v2.0)[108]. **b** Western blot of the anti-Flag co-immunoprecipitation of the described VHF-tagged FRQ strains. Results are representative of three biological replicates from four different experiments (one biological replicate has two technical replicates including anti-Flag and anti-V5 pull-downs), shown in Supplementary Fig. 5a. * here denotes a non-specific band in the Anti-FRH Input lanes. **c** Densitometry analysis of western blots on FRQ strains treated with cycloheximide (CHX). A two-way ANOVA for relative FRQ levels based on strain and time was significant (Strain factor $F = 13.88$, $p < 0.0001$; Time factor $F = 32.25$, $p < 0.0001$). Tukey's multiple comparison-adjusted *p*-values are shown within each time point (*$p < 0.05$, ***$p < 0.001$, ns not significant). **d** Densitometry analysis of western blots for relative levels of WC-1 among the different strains considered. One-way ANOVA comparing relative WC-1 levels between strains was significant ($F = 20.09$, $p < 0.0001$), with significant Tukey's multiple comparisons test adjusted *p*-values shown (*$p < 0.05$, **$p < 0.01$, ***$p < 0.001$, ****$p < 0.0001$). Error bars in (**c**) and (**d**) denote standard deviation, $n = 3$ biological replicates from one experimental round, exa**ct** *p*-values are available in the Source Data file. Note that due to an air-bubble overlapping the FRQ band, FRQ^VHF only has an $n = 2$ at 4 h post-cycloheximide. FRQ (FREQUENCY), FRH (FRQ-INTERACTING RNA HELICASE), WC-1 (WHITE COLLAR-1). Source data are provided as a Source Data file.

## Discussion

A mechanistic understanding of how the circadian TTFL regulates biology has been hampered by the highly disordered nature of some core clock proteins. Our application of the LOCATE approach allowed us to overcome this hurdle by granting high amino acid resolution analysis of how FRH binds to different peptides representing segments of FRQ. This analysis identified essential double arginine residues just beyond the previously identified FFD SLiM within FRQ that were important for binding with FRH (Fig. 5b). This expanded FFD SLiM highlights that our method was able to both recapitulate but also update the sequence-specific SLiM that is essential for the interaction between FRQ and FRH. Unexpectedly, the RR783AA mutation stabilized FRQ (Fig. 5c). While the precise molecular mechanism remains untested, the most parsimonious explanation is that the RR783AA mutations affected a local degron within FRQ that is usually shielded when its FFD region interacts with FRH. This explanation is supported by the prediction of a degron immediately N-terminal of the FFD region of FRQ, though this remains to be investigated. This also offers a potential mechanism through which FRH could nanny FRQ: upon FRH binding via the FFD, the putative degron would be hidden and prevent

FRQ degradation. This is further in keeping with other recent work highlighting how degrons are often masked by interactors to regulate protein stability[55].

The stabilizing RR783AA mutation allowed us to investigate the role of FRQ in the core clock independently of FRH. Our work demonstrated that FRQ was able to exert negative feedback onto the WCC without FRH, to create an hourglass-like circuit (Fig. 6b–d). In support of our model that FRQ can repress the core clock in the absence of FRH, previous work on FRQ has shown that it can enter the nucleus without FRH and also interact with the WCC and its principle kinase Casein Kinase 1a (CK-1A) independently of FRH[18,56,57]. In higher eukaryotes, CRYPTOCHROME (CRY) has also been shown to be involved in persistent rhythms[58], but the mechanism by which this persistence is imparted by either CRY, or in this case FRH, is unclear and of high interest for future work. While it is possible that the dampening seen within the FRQ^RR/AA strain in Fig. 6b could be due to desynchronization amongst its cellular clocks, the clear banding that occurs more quickly after the light-dark transition in FRQ^RR/AA compared to FRQ^VHF or FRQ^RR/HH in Fig. 6c suggests that more rapid feedback is occurring in this strain that could dampen all its cellular TTFLs

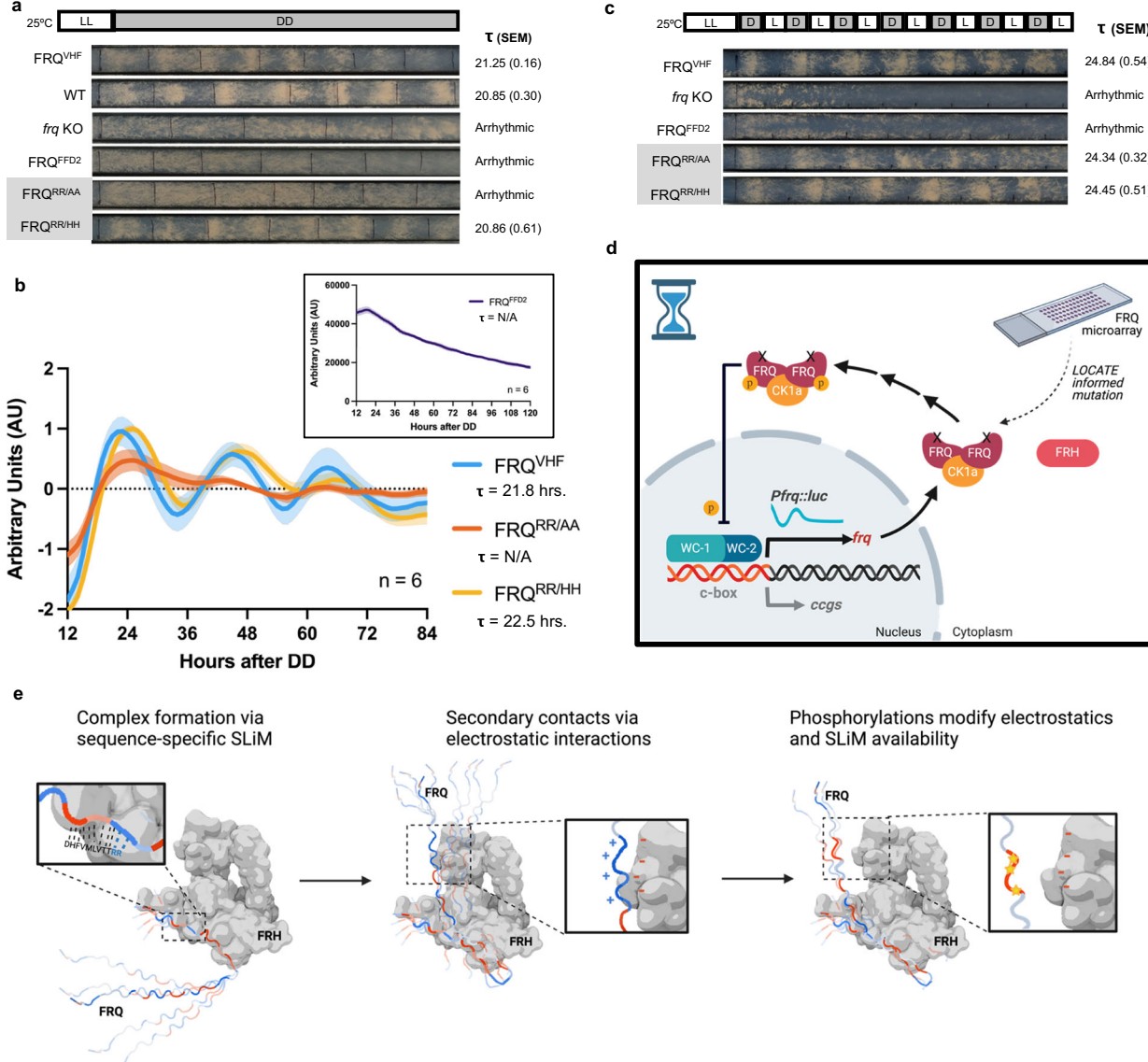

**Fig. 6 | Loss of FRQ-FRH interaction leads to loss of clock persistence but not feedback. a** Representative race tubes of the VHF-tagged FRQ strains grown in constant darkness (DD). Average period (τ) in hours (with SEM) derived from $n = 6$ race tubes. See Supplementary Fig. 7a for images of all race tubes. **b** Smoothed, linear detrended and normalized average and standard deviation of luciferase levels over 3 circadian days of the VHF-tagged FRQ strains with a luciferase reporter for *frq* promoter activity (*frq*$_{c\text{-}box}$-*luc*[+]) ($n = 6$) grown in DD in a 96-well format in the presence of luciferin. Inset of smoothed FRQ[FFD2] is shown for comparison. See Supplementary Fig. 6c for other FRQ[RR/AA] biological replicates. **c** Representative race tubes of FRQ[VHF] ($n = 5$), *frq* KO ($n = 5$), FRQ[FFD2] ($n = 6$), FRQ[RR/AA] ($n = 5$), and FRQ[RR/HH] ($n = 6$) strains grown under a 12L:12D lighting regime in one experimental round. See Supplementary Fig. 7b for all race tubes. **d** Hourglass model based on our LOCATE-informed mutation FRQ[RR/AA]. In this strain, FRQ is not able to interact with FRH yet can feed back on WC-1/2 to close the circuit in an hourglass-like manner. The loss of persistence in the clock is demonstrated by the damping

oscillation of luciferase as expressed from the *frq* promoter. **e** Illustration of the multivalent, conformationally heterogeneous interaction model for FRQ and FRH. The FRQ/FRH complex forms primarily via the sequence-specific SLiM including the additional essential RR residues. Secondary contacts could then form between positively charged clusters within FRQ and the overall negative outer surface of FRH, yielding a dynamic complex that samples different conformations. Finally, phosphorylation of FRQ (yellow stars) could modify charge patterning and therefore change interactions with FRH and potentially other interactors via changes in SLiM availability. The illustration uses a space-filling model of a known crystal structure of FRH (PDB5E02; gray protein), but the linear representation of FRQ is not shown to scale (modified from a 50 a.a. segment of FRQ modeled in AlphaFold2, see "Methods" section). Blue regions on the linear FRQ protein represent positive charge clusters and red regions represent negative charge clusters. Abbreviations as in Fig. 1. Images in (**d**) and (**e**) were created with BioRender.com. Source data are provided as a Source Data file and at [https://doi.org/10.17632/7hgspb5gn7.1].

(Fig. 6c). Further, as *N. crassa* exists as a syncytium of interconnected multinucleated cells, this suggests the likelihood of rapid desynchronization is less likely than dampening of the intracellular core clock[59].

Beyond the essential SLiM that allows FRH to bind FRQ and create persistent oscillations from the TTFL, LOCATE demonstrated that FRQ peptides with a positive net charge per residue bound FRH (Fig. 3a). This was possible to uncover due to the ability of immobilized peptide microarrays to capture a wide range of interaction affinities, including

"weaker" interactions[13]. Ablating one such cluster of positive charges from FRQ revealed its importance for proper clock function in vivo. Specifically, mutation of a defined positive charge block (KKK315AAA) led to a lengthening of the core clock period and the loss of overt clock output (Fig. 3d, e)[22]. In previous work examining transcriptional regulators, charge blocks were found to be essential for proper protein localization and interaction[22]. Therefore, we interpret these observed changes, along with the observation of FRH binding to positively

charged peptides of FRQ, to be due to the modification of the local electrostatic interaction between this positive charge block and FRH that could impact downstream interactions, though this still requires follow-up studies to confirm. The LOCATE data, in combination with a lack of change in the interaction between FRQ and FRH when a positive charge block is mutated, is consistent with the positive charge blocks mediating multivalent FRQ/FRH interactions at secondary sites. This effect could also be due to a change in FRQ interaction with other proteins. However, as opposed to other studies where lengthened clock periods or overt arrhythmicity are related to changes in FRQ interaction with CK-1A[34,39,60,61], the ~35 h period of our FRQ^KKK/AAA mutant is surprisingly not related to an observable change in binding with CK-1A (Supplementary Fig. 3d, e). Future studies will need to identify which protein interactions may have been affected in the FRQ^KKK/AAA mutant.

Further, our analysis highlighted that positively charged clusters were a consistent molecular feature across FRQ orthologs in fungi and higher eukaryotes (Fig. 3c and Supplementary Fig. 2c)[20,33]. Given our results, we propose a model in which disordered negative arm proteins (FRQ/ the PERs) bind to a partner protein (e.g., FRH/ the CRYs) through sequence-specific interactions (Fig. 6e). Once bound, the positively charged clusters along the length of FRQ or the PERs would bind their partner protein at secondary sites to generate a multivalent and likely dynamic protein complex with a range of conformations that present or shield binding motifs on the disordered negative arm proteins (Fig. 6e)[15,49,62]. In keeping with this theory, predicted SLiMs for downstream interactors tend to occur in regions of FRQ that became negatively charged over the circadian day due to phosphorylation (Supplementary Fig. 2e, f). The implication is that daily phosphorylation systematically modulates charge blocks, changing conformations that affect interactions between FRQ and FRH (and other interactors) and temporally rewiring the FRQ interactome (Fig. 6e)[15]. This is supported by the suggestion of different conformations of FRQ over the circadian day as evidenced by differential protease accessibility that are correlated with the timing of different circadian-centered macromolecular complexes that control periodicity and output[15,20,63].

One caveat is that our LOCATE assay enables us to assess local binding propensity but does not allow us to assess the dynamic and cooperative nature of potential multivalent binding or measure the exact $K_d$ of any interactions. To confirm that FRQ/FRH form a dynamic bound-state complex would necessitate a structural and/or conformational assessment of the bound state, as has been done in other contexts[64–67]. While major progress has been made in performing integrative biophysical dissection of dynamic protein complexes, these approaches are often technically demanding (e.g., ref. 63). Rather, our LOCATE assay instead offers an accessible and high-throughput method to identify and study IDRs of interest, complementary to what can be done with structural biology. Another limitation of our study is that the first 100 a.a. of FRH were not present in our construct, therefore further binding could occur between these first 100 a.a. residues of FRH and the FRQ peptides. We also do not have reciprocal information about what regions of FRH are binding to these peptides, and there remains much work to be done to understand FRH/FRQ binding from the perspective of the FRH protein. Planned follow-up work will include SPR between FRH and different combinatorial mutants of FRQ to better understand the potential effects of FRQ's positive charge blocks on the FRQ/FRH complex. A final caveat is that we did not screen our FRQ microarray with an empty Histidine epitope tag and cannot eliminate the possibility that, although histidine is slightly protonated at pH 7.0, the His-tag could have affected binding.

Molecular interactions mediated by intrinsically disordered regions have historically been viewed through the lens of sequence-specific molecular recognition driven by short linear motifs (SLiMs)[8]. Our work supports this model by identifying specific binding motifs

and demonstrating that LOCATE can be used as an in vitro high-throughput tool to characterize these interactions. Beyond SLiMs, a second, complementary mode of IDR-mediated interaction termed chemical specificity has been proposed[68]. Here, complementary chemical interactions between an IDR and its binding partner do not require a specific amino acid sequence but rely on the presence of certain chemical moieties. Our work here adds to a growing number of examples in which clusters of charged residues offer one type of chemical specificity that can mediate intra- and inter-molecular interactions[22–24,29,69]. Importantly, our work demonstrates the functional impact of altering charge blocks in vivo, and the usefulness of this framework has been supported by other more recent work showing that clusters of negatively charged residues within FRQ mediate its electrostatic interaction with proteins in the positive arm of the clock (WCC)[70]. Given that phosphorylation alters the balance of positive and negative charge blocks over the circadian day (Supplementary Fig. 2e), how a changing electrostatic profile may drive positive-arm repression will also be interesting to uncover in future studies.

Taken together, our results expand our growing appreciation for how seemingly poorly conserved intrinsically disordered regions can drive core cellular functions. An emerging paradigm suggests that charge clustering contributes to the regulation of many different protein-protein interactions in the cell, which now includes some of the proteins that participate in the circadian clock. Given that the core clock must be responsive to changes in cellular state from many different signaling pathways, protein-protein interactions mediated by clusters of charged residues could offer an additional layer of semi-specific molecular recognition, enabling these clock proteins to participate in a wide array of different protein interactions. We tentatively suggest this may offer one possible reason for the prevalence of intrinsically disordered regions and charge blocks in the core clock proteins.

## Methods

### Peptide library design, synthesis, and microarray printing
Two peptide libraries were designed, with Library I primarily made up of a linear peptide "mapping" of the primary sequence of FRQ (NCU02265) and Library II containing further rationally designed peptides to investigate the specificity of FRH binding to original "parent" peptides (see Library details at https://doi.org/10.17632/7hgspb5gn7.1). The FRQ "mapping" peptides in Library I were 15 amino acids (a.a.) in length, beginning at the N-terminus of FRQ's sequence, with consecutive peptides shifting by 3 a.a. Peptide mapping for some regions of interest (e.g., FFD region) were repeated at a finer scale of 15 a.a., shifting by 1 a.a. between consecutive peptides. Any parent peptides of interest were used as the basis of further rationally designed peptides, such as scrambled sequences (using the Genscript random library tool; https://www.genscript.com/random_library.html) or truncation series, to identify sequence-specific interactions and the minimal SLiM. Parent peptides containing a candidate SLiM were further singly or doubly mutated to other residues of interest (not all combinatorial possibilities were investigated) to identify permissive or prohibited residues at different positions within the SLiM.

Peptides were synthesized using standard Fluoroenylmethyloxycarbonyl (Fmoc) chemistry in an automated peptide synthesizer (Multipep RS, INTAVIS Bioanalytical Instruments AG, Germany), as done previously[17]. Briefly, parallel peptide synthesis proceeded from C- to N-terminus on solid cellulose disks, with all peptides N-terminally acetylated to better mimic the charge of a peptide segment within the parent protein's sequence. Note that the first peptide based on the parent protein's N-terminus was not N-terminally acetylated. Once synthesized, peptide-cellulose disks were reconstituted in 250 μL dimethyl sulfoxide (DMSO) following Intavis' standard work-up

procedure. The resulting peptide stock solutions were used 1:1 for spotting in triplicate in a microarray format on nitrocellulose-coated glass microscope slides using a slide-spotting robot (Intavis Bioanalytical Instruments AG). Peptide microarrays were then air-dried for 2 h at 65 °C.

## FRH protein expression and purification

The FRHΔ100 plasmid[21], consisting of a pET28a vector, FRH (100-1106 a.a.), and an N-terminal 6× His-tag, was transformed into BL21 (DE3) Competent *E. coli* cells (New England Biolabs, C2527) and plated on selective LB plates containing 30 μg/mL kanamycin (AMRESCO, 0408-10G). Positive colonies were grown in 3 L of liquid LB with 30 μg/mL kanamycin at 37 °C, shaking at 225 rpm, until $Abs_{600}$ was ~0.45. After cooling the culture on ice, protein expression of the FRHΔ100 plasmid was induced with 0.2 mM IPTG (Biotium, 10021), at 18 °C, shaking at 185 rpm, for 16 h. Cells were pelleted by centrifugation at 4 °C, 2833 rcf, for 15 min, and kept on ice for immediate protein extraction. Protein extraction buffers contained 50 mM HEPES (Sigma Aldrich, H0887-100 mL) and 150 mM NaCl, pH 7.0, 2 mM beta-mercaptoethanol, and varying amounts of imidazole as outlined below (Amresco, 0527-100G). Pelleted cells were resuspended in 30 mL of Lysis Buffer that contained 10 mM imidazole. Cells were lysed by three rounds of French Pressing at 1000 psi. The cell lysate was clarified by centrifugation at 4 °C, 20,817 rcf for 20 min. The soluble fraction was split between two columns each with 2 mL Ni-NTA agarose beads (Qiagen, 30210) pre-equilibrated with Lysis Buffer, and nutated for 1 h at 4 °C. The columns were washed with two rounds of 15 mL Wash Buffer containing 30 mM Imidazole. Proteins were eluted in 500 μL fractions using an Elution Buffer with 200 mM imidazole. Each elution was applied to a 40 kDa cut-off Zeba column (ThermoScientific, 87769) to de-salt and exchange the buffer to PBS (100 mM NaCl, 10 mM Potassium Phosphate, pH 7.0). A BSA-based Bradford assay (Bio-Rad, 5000006) was used to quantify resulting amounts of FRH protein, and the percentage of FRH in the final product was visually estimated based on the percentage of the FRH band relative to other bands (~30% FRH) when the eluted protein was visualized on a Coomassie-stained gel (Amresco, 0472-25G). Expression and purification of FRHΔ100 were verified by SDS-PAGE analysis (ThermoScientific, WG1602BOX), and Western blotting with primary anti-His (clone 6AT18, mouse) at 1:1000 (Sigma Aldrich, SAB1305538), secondary Goat anti-Mouse polyclonal at 1:10,000 (ThermoFisher, 31430) and PICO (Thermo-Scientific, 34577). Further purification was not considered as the non-specific protein products acted beneficially as a built-in competition assay to decrease false positives for FRH binding.

## Microarray incubation and data analysis

Basic microarray screening protocol was carried out at room temperature on a rocker using 5 mL of each buffer per slide, as in ref. 17. FRQ-based microarrays were first blocked for 3 h in PBS (10 mM phosphate, 100 mM NaCl, pH 7.0) with 5% w/v BSA, followed by 3 × 10 min washes in PBS. Incubation with different approximate concentrations (1, 10, or 100 nM) of the purified and buffer-exchanged FRH occurred for 3 h followed by another 3 × 10 min washes and 1 h incubation with antibodies (in 2.5% w/v BSA in PBS) with 3 × 10 min washes between each step, either anti-His at 1:1000 then anti-Mouse at 1:500,000 (see above), or anti-FRH polyclonal (rabbit) at 1:12,500[71] followed by Goat anti-Rabbit polyclonal at 1:500,000 (Invitrogen, 31460), then SuperSignal West FEMTO (ThermoScientific, 34094). Non-specific antibody binding was assessed by incubating with antibodies and FEMTO, but without first incubating with FRH. Chemiluminescence imaging was performed with a ChemiDoc XRS+ System (Bio-Rad) and Image Lab 4.0 software, using signal accumulation mode (SAM) with the high-resolution option with 2 × 2 binning.

Microarray images were normalized for each library within Image Lab by standardizing the intensity range to allow better comparison amongst replicates, and exported to Fiji (ImageJ v2.0.0, NIH) to convert to 8-bit grayscale, inverted, and background subtracted (rolling ball radius = 50 pixels)[72]. Merged spots, missing spots, or misprinted aberrantly shaped spots were manually excluded only when necessary (affecting no more than 10 spots out of 1152 per slide and sometimes no spots excluded). TIGR Spotfinder (Release 2009-08-21) was used to quantify remaining spot intensities using the Otsu segmentation method with local background subtraction[73]. Undetected spots within TIGR Spotfinder had a value of 0 as they were indistinguishable from background intensities. Extracted intensity data are available at [https://doi.org/10.17632/7hgspb5gn7.1]. Basic plots of sequential normalized microarray intensity, truncations, and scrambled peptides were all plotted in PRISM 9.0.2. Compositional analysis of the peptide residues was also carried out as in Shastry & Karande[17], and plotted in PRISM 9.0.2. Briefly, the frequency of occurrence of residue types (acidic, basic, polar, aromatic, or nonpolar) in the 3-mer shift peptide mapping portion of the peptide library were compared to the frequency of occurrence in the top 10% of peptides, i.e., the peptides with the highest normalized binding intensities. The statistical significance of the changes in residue occurrence amongst populations was determined using the test statistic z:

$$z = \frac{\hat{p} - p_0}{\sqrt{\frac{p_0(1-p_0)}{n}}} \tag{1}$$

where $\hat{p}$ is the percentage of a residue or residue type occurring in the top 10% of peptides; $p_0$ is the percentage of the same residue or residue type occurring in the peptide mapping portion of the library; and $n$ is the total cumulative number of residues in both the partial peptide mapping library and the top 10%. For net charge at pH 7.0 calculations, residues D and E were each considered -1 charge, R and K as +1 charge, and H as +0.091, while all other residues were considered charge neutral. The resulting peptide charges were plotted in PRISM v9.0.2, along with a densitometric plot produced in R v4.3.1. Analysis for the FFD region was calculated as a fraction of the WT peptide LPDDHFVMLVTTRRV value of 64, an average of the Library II peptides 215 and 235 (see https://doi.org/10.17632/7hgspb5gn7.1 for more details), that was then transformed to the $\log_2$ scale.

## Electrostatic potential calculation for FRH

The solved crystal structure of FRH, PDB5E02, was downloaded from the Protein Data Bank[21]. Using PyMol (v2.4.0), the file was converted to PQR format and the APBS tool used to calculate the solvent-accessible electrostatic potential according to the Poisson-Boltzmann equation. The electrostatic map was visualized in PyMol, using a gradient of −3 kT/e (red) to +3 kT/e (blue).

## Bioinformatics for BLAST multiple sequence alignment and cladogram

The *N. crassa* FREQUENCY (NCU02265) protein sequence was run in NCBI BLAST[74]. Homologous protein sequences with over 50% identities or 50% positives with FRQ were chosen for further analysis, yielding a list of ten other homologous proteins. Fungal protein sequences of interest were exported from FungiDB, release 47[75]. Next, the protein sequences were aligned using the UniProt alignment tool and exported in Stockholm format[76]. The multi-sequence alignment was further visualized in SnapGene (version 5.1.1) and a logo was made using the interactive tool Skylign (accessed July 2020). Parameters used in Skylign were logos based on the full alignments, letter heights determined by information content above the background, and their weighted counts method, where weights are applied to account for highly similar sequences before calculating a maximum-likelihood estimate for each column in a multi-sequence alignment[77]. The relationship amongst the species included in our FRQ homology analysis

was represented in a rectangular cladogram, created using the Interactive Tree of Life online tool, ver. 4[78].

## Sequence analysis

Overall predictions of FRQ sequence disorder were carried out using the package metapredict (v1), a machine learning-based method that predicts whether a residue is in a disordered region by predicting that residue's consensus score (smoothing using a 5 residue window) across multiple disorder predictors[79]. Metapredict considers residues with a predicted consensus value of >0.3 as disordered (meaning that more than 30% of the predictors agreed that the residue was in a disordered region), while residues <0.3 are considered ordered. To calculate sequence conservation, we extracted 86 orthologs N. crassa FRQ sequences from eggNOG (v5.0)[80], aligned these sequences, and then calculated per residue conservation scores by using the approach as described by Capra et al.[81], grouping amino acids by residue properties. Using the primary amino acid sequence of FRQ (NCU02265), we calculated the net charge per residue (NCPR) using a sliding window of 10 residues and average Kyte-Doolittle hydropathy using the localCIDER program (v0.1.18) with a sliding window size of 5, meaning the value at each position is based on that residue and the two residues to each side[82]. NCPR was also recalculated at different timepoints including the phosphosites reported in Baker et al.[20] as negative charges (each phosphorylation was considered -2). We measured the clustering of particular residue classes (e.g., positive, negative, and aromatic residues) within FRQ's sequence by calculating the average inverse-weighted distance (IWD). The IWD is defined as:

$$IWD = <\frac{1}{d}> = \frac{1}{N_{pairs}} \sum_{i=1}^{N_S-1} \sum_{j=i+1}^{N_S} \frac{1}{S_j - S_i} \qquad (2)$$

where $S$ is the set of target residue positions, $S_i$ is the $i$th element of $S$, $N_S$ is the number of items in $S$, and $N_{pairs}$ is the number of pairwise combinations between elements of $S$[83,84]. Following the approach presented in ref. 31, IWD was compared to a null distribution of IWD values calculated from 10,000 randomly shuffled FRQ sequences. Significance was assessed as IWD values that fall outside of the 95% confidence interval of this null distribution. To facilitate comparisons across orthologs that differ in length and amino acid composition (both of which affect IWD), we normalized IWD to a Z-score. IWD was also calculated for a set of N. crassa proteins ($n = 1295$) as follows: all Neurospora proteins were accessed from Uniprot (accessed Jan. 2023) and filtered for the GO terms "cytoplasm" [GO:0005737] or "nucleus" [GO:0005634]. In keeping with other recently published work, we also define biologically relevant charge "blocks" according to Lyons et al.[22]. Using a scanning window of 10 a.a. and merging sets of windows (peptides) sharing a positive NCPR where at least one window has NCPR >= 0.5 would be a positive charge block, while at least one 10 a.a. segment in a negative charge block would have NCPR <= −0.5 to be considered a negative charge block.

## All-atom simulations

All-atom simulations were performed using the CAMPARI simulation engine (v3) with the ABSINTH implicit solvent model[85,86] (http://campari.sourceforge.net/). A 50-residue fragment from FRQ (a.a. 299–348) including the FCD-1 region (Fig. 1b) was simulated in a spherical droplet with a radius of 94 Å and compared to a KKK315AAA construct, as has been done previously[27,87,88]. Briefly, simulations were performed at 340 K and 15 mM NaCl. Ten independent simulations were run for each of the different constructs. Each simulation was run for $126 \times 10^6$ Monte Carlo steps, with the first $6 \times 10^6$ discarded as equilibration. Trajectory information was saved every 80,000 steps, such that the final ensembles consist of 15,000 distinct conformations. This was repeated for a 50-residue fragment from FRQ (a.a. 754–803).

and compared to the RR783AA and RR783HH constructs with both neutral histidine, and positively charged histidine.

Simulations were analyzed using MDTraj and SOURSOP v0.2.0 (https://soursop.readthedocs.io/)[89]. The secondary structure was calculated using the DSSP algorithm[90]. The solvent-accessible surface area was calculated for the sidechains only, using a probe radius of 1.4 Å. To compare changes in the solvent-accessible surface area (SASA) along the sequence necessitates correcting for the intrinsic differences in sidechain volume. To account for this, we performed simulations in which all attractive non-bonded interactions are turned off. In these simulations, only the repulsive component of the Lennard–Jones potential determines the energetically accessible ensemble. This excluded volume (EV) ensemble allows for the intrinsic SASA of each reside sidechain in the appropriate sequence context to be computed, as done previously[91]. The normalized SASA is then calculated as the ratio of the per-residue SASA from the full simulation divided by the SASA from the EV ensemble. Finally, the change in normalized SASA was computed by calculating the difference between the normalized SASA in the wild-type simulation and each of the variants. The Analytical Flory Random Coil (AFRC) model was used to contextualize the radius of gyration distribution in the Supplementary Figs. 3 and 5[92].

## Generation of FRQ^VHF and FRQ mutant strains

An FRQ^VHF cassette was developed for insertion into the cyclophilin locus of N. crassa as described[35]. Starting at the 5′ end, we fused 1000 bp from the upstream portion of the cyclophilin locus (csr-1, NCU00726) from FungiDB to the 3000 bp upstream promoter of the frq ORF and the complete wild-type frq Open-Reading Frame (ORF), followed by a 10× glycine linker and V5-10-His-3-Flag (VHF) tag[36,37], followed by a new stop codon, followed by 1000 bp downstream of the frq ORF, and finally 1000 bp from the downstream portion of the target csr-1 locus[75]. Template genomic DNA was harvested from a wild-type N. crassa strain 87-3 (ras-1^bd, mat a) using the Gentra Puregene Tissue kit (Qiagen, 158622) and following the manufacturer's instructions. PCR reactions using primers found in Table 1 were carried out using Phusion Flash High-Fidelity PCR Master Mix (ThermoFisher, F548S), following the manufacturer's instructions and using an Eppendorf Mastercycler Nexus Thermal Cycler with the following program: 98 °C 10 s, (98 °C 5 s, 67 °C 5 s, 72 °C 4 min) ×35, 72 °C 5 min, 4 °C 5 min. Appropriate cDNA product sizes were confirmed using a 0.8% agarose gel in 1× TAE buffer, relative to a 1 kb DNA ladder (New England Bio, N3232L). The cDNA pieces with primer overhangs were ligated along with a gapped selective yeast vector (pRS426, containing URA3 and ampicillin resistance; Hurley et al.[19]) using the homologous recombination system endogenously found in yeast (strain FY834 from the Fungal Genetics Stock Center), by carrying out a Lithium Acetate/ PEG transformation[93,94]. After isolating all plasmid DNA using a "Smash and Grab" protocol[95], the harvested plasmids were transformed into E. coli (DH5alpha derivative; New England Bio, C2989K) using the manufacturer's instructions (BTX Harvard Apparatus, 45-2001) and plated onto LB agar plates with 100 μg/mL ampicillin to grow overnight at 37 °C. Colonies were picked and grown in liquid LB broth with 100 μg/mL ampicillin overnight at 37 °C and shaken at 225 rpm. After centrifugation of the culture, plasmids were isolated and purified using the QIAprep Spin Miniprep kit (Qiagen, 27106), and PCR of the final linear cassette was done as stated above except using primers MSJ001F and MSJ006R. The cassette was purified using the QIAquick PCR Purification Kit (Qiagen, 28106). Cassette sequences were verified through Sanger sequencing using sequencing primers spaced ~700–800 nt apart (Genewiz and Eurofins Operon, LLC). Verified cassettes were transformed into the csr-1 locus of an frq KO N. crassa strain, 122 (delta-frq::hph+, ras-1^bd, mat a), using methods previously described[35]. Transformants recovered after electroporation in a liquid VM medium with additional yeast extract and were then plated

**Table 1 | Primers used in this study synthesized from Integrated DNA Technologies, Inc. IDT**

| Purpose | Primer Name | Forward or Reverse Complement? | Primer Sequence (mutation underlined) |
|---|---|---|---|
| Joining plasmid pRS426 to 5′ csr-1 locus | MSJ001F | Forward | GGGTTTTCCCAGTCACGACGGGGTCTGCAGCTGTACCGGG |
| FRQ^VHF; 5′ csr-1 to frq locus | MSJ002F | Forward | GAACCGTGCTTAATCAGGTACGGAAGAGGTTGTTGCGAACAAAG |
| FRQ^VHF; 5′ csr-1 to frq locus | MSJ002R | Reverse Complement | CTTTGTTCGCAACAACCTCTTCCGTACCTGATTAAGCACGGTTC |
| FRQ^VHF; frq ORF to VHF tag | MSJ003F | Forward | GATGGAGGACGTCTCATCCTCGGGCGGAGGCGGCGGGAGGCGG |
| FRQ^VHF; frq ORF to VHF tag | MSJ003R | Reverse Complement | CCGCCTCCGCCGCCTCCGCCCGAGGATGAGACGTCCTCCATC |
| VHF tag template (Glycine lin-ker,3-Flag/10-His/V5 tag). | MSJTag001 | Forward | GGCGGAGGCGGCGGAGGCGGAGGCGGAGGCGGGCGGTAAGCCTATCCCTAACCCTCTCCTCGGGTCGGATTCTACG/CATCATCACCATCACCATCAT-CACCACCAC/GACTACAAAGACCATGACGGTGATTATAAAGATCATGACACTGACTACAAGGATGACGATGACAAGTAG |
| VHF tag to 3′ end of frq locus | MSJ004F | Forward | CAAGGATGACGATGACAAGTAGGACCTGAGTGGTATTTTC |
| VHF tag to 3′ end of frq locus | MSJ004R | Reverse Complement | GAAAAATACCCACTCAGGTCCTACTTGTCATCGTCATCCTTG |
| frq locus to 3′ csr-1 locus | MSJ005F | Forward | GGGCGGGCTACACAGACAGTCAACGCCTAGATGAAACCAAATTAC |
| frq locus to 3′ csr-1 locus | MSJ005R | Reverse Complement | GTAATTTGGTTTCATCTAGGCGTTGACTGTCTGTGTAGCCCGCCC |
| Joining plasmid pRS426 to 3′ csr-1 locus | MSJ006R | Reverse Complement | CAATTTCACACAGGAAACAGCGCCGACTCGCTTATGAAGCATTG |
| FRQ^KKK/AAA; KKK315AAA | MSJ009F | Forward | CATATTGTCATGACCGACCAAGGAGGCCGCTGCCCTGGTTGTCCGACGCTTG |
| FRQ^KKK/AAA; KKK315AAA | MSJ009R | Reverse Complement | CAAGCGTCGGACAACCAGGGCAGCGGCCTCCTTGTCGGTCATGACAATATG |
| FRQ^FFD2; VMLVTT777AAAAAA | MSJ011F | Forward | CTTCCTGACGACCATTTTGCTGCCGCCGCGGCGGCGCGCGTCGTCGAGACCTATC |
| FRQ^FFD2; VMLVTT777AAAAAA | MSJ011R | Reverse Complement | GATAGGTCTGACGACCGCGGCGGCGGCGGCGGCGCAGCAAAATGGTCGTCAGGAAG |
| FRQ^RR/AA; RR783AA | MSJ012F | Forward | GTGATGCTCGTCACCACTGCCGCCGTCGTCAGACCTATCCTG |
| FRQ^RR/AA; RR783AA | MSJ012R | Reverse Complement | CAGGATAGGTCTGACGACGGCGGCAGTGGTGACGAGCATCAC |
| FRQ^RR/HH; RR783HH | MSJ013F | Forward | GTGATGCTCGTCACCACTCACCACGTGCGTCAGACCTATCCTG |
| FRQ^RR/HH; RR783HH | MSJ013R | Reverse Complement | CAGGATAGGTCTGACGACGTGGTGAGTGGTGACGAGCATCAC |

Underlined bases denote the substitutions made for the different FRQ mutations. Bases in italics relate to the latter portion of a forward primer that matches the second listed gene locus under "Purpose".

on an agar plate containing a growth-restrictive mixture of fructose-glucose-sorbose (FGS) and 5 µg/mL of cyclosporin A[35,96]. After 3-4 days of growth at 25 °C, isolated colonies were picked and placed onto selective slants (small test tubes with agar-based VM media) and 5 µg/mL cyclosporin A (Sigma, 20024).

We designed our FRQ mutations around the concept of maintaining the relative adaptiveness of codons used, since previous studies have shown that FRQ codon usage has effects on its structure and stability[97]. By accessing the codon table for *N. crassa*[98], we calculated the relative adaptiveness of each codon, with 100% being the highest frequency amino acid codon, and then relatively scaled the frequency of the remaining codons[99]. We used this Relative Adaptiveness measure of codons to rank from most adaptive to least adaptive and substituted an alanine or histidine codon of similar rank for our mutations. For our FRQ-FFD mutants, the program PrimerX (http://www.bioinformatics.org/primerx/cgi-bin/DNA_1.cgi) was used to design the needed primers, using the following specifications: Melting temp. 50–85 °C, GC content 40–60%, length 40–60 bp, 5′ flanking region 20–30 bp, 3′ flanking region 20–30 bp, terminates in G or C, mutation site at center, and complementary primer pair[100]. Designed primers are found in Table 1.

For luciferase reporter assays, we created a new strain (uber #6) which was the result of a sexual cross between the X200-3 strain (*frq*⁺, *his-3::frq_{c-box}-luc*⁺, *ras-1^{bd}, mat A*) and the 122 strain (*delta-frq::hph*⁺, *ras-1^{bd}, mat a*), to create an *frq* KO strain with a luciferase reporter attached to a minimal *frq* promoter in the background (*delta-frq::hph*⁺, *his-3::frq_{c-box}-luc*⁺, *ras-1^{bd}*). All cassettes created above were then re-transformed into the *csr-1* locus of this new *frq* KO strain with luciferase reporter (uber #6). For the race tube and Co-IP assays, we used WT strain 328-4 (*frq*⁺, *ras-1^{bd}, mat A*).

To perform the Co-IP assay to investigate FRQ^{KKK/AAA} interactions with FRH and CK1a, we carried out a further sexual cross between uber #6 (*delta-frq::hph*⁺, *his-3::frq_{c-box}-luc*⁺, *ras-1^{bd}, mat A*) and 811-2 (*frqQ494A,L495A,H496A^{VSH6}::hph*⁺, *short-ck1a^{HA}::bar*⁺, *ras-1^{bd}, mat a*) (courtesy of Chris Baker and the Dunlap-Loros labs[101]). Successful progeny were identified by western blot, to produce a strain with *delta-frq::hph*⁺, *short-ck1a^{HA}::bar*⁺, *ras-1^{bd}*. Our FRQ^{VHF} and FRQ^{KKK315AAA} constructs were then re-transformed into the *csr-1* locus of this new cross, as above.

All strains are available upon request and will be deposited at the Fungal Genetics Stock Center (FGSC).

## Race tube assays and period determination

Race tube assays were carried out using custom glass tubes filled with 14 mL of race tube medium (1× Vogel's salts, 0.05% glucose, 0.1% arginine, 50 ng/mL biotin, and 1.5% bacto-agar). See fgsc.net/neurosporaportocols/How to choose and prepare media.pdf for more details on Vogel's salts or other Neurospora media. Prepared race tubes were inoculated with 10 µL of a conidial suspension from the noted strain and grown for ~24 h at 25 °C in constant light (LL), before being synchronized with a transition to constant dark (DD), 25 °C, and marked each 24 h until strains reached the end of the race tube. Race tubes were scanned using an EPSON GT-1500 scanner, and images were cropped and converted to black and white. Clock period was analyzed using ChronOSX (v1.1.0)[102], using the Period Analysis option and including 6 days of constant dark densitometry data from each race tube to calculate the mean and standard deviation for each tube, followed by a mean and standard error of the mean for each set of replicates per strain.

## CCD array trials and analysis

Low Nitrogen-CCD media (LN-CCD; 0.03% glucose, 0.05% arginine, 50 ng/mL biotin, 1× Vogel's salts, 1.5% bacto-agar, 25 µM luciferin) with 0.001 M Quinic Acid (pH 4.75) was used for all luciferase reporter assays. 185 µL of this media was used per well in a white 96-well plate

(Greiner Bio-One, 655073), inoculated with 10 µL of the relevant conidial suspension. Plates were sealed with a breathable membrane (BreatheEasy, USA Scientific, 9123-6100), and incubated at 25 °C LL for ~48 h before being placed at 25 °C DD in an incubator with a PIXIS CCD array (Princeton Instruments, 1024B) with a 35 mm Nikon DX lens (AF-S NIKKOR, 1:1 8G), run by the program Lightfield (version 5.2, Princeton Instruments). Images were acquired for 15 min every hour and final image stacks were imported into FIJI (ImageJ v2.0.0, NIH) to adjust brightness, and denoised using the default "Remove outliers" tool option. A custom image analysis plugin was used called "Toolset Image Analysis Larrondo's Lab 1.0" (courtesy of Luis Larrondo) using the 96-well plate quantifying tool. Wells that had less media, low growth, and low luminescence levels or luminescence that did not continue above the background until the end of the trial were not considered further. Of the remaining wells, all luminescence traces were similar in pattern, but the six most similar in absolute intensities were used for further analysis to allow optimal model fitting. The extracted luciferase reporter data were then analyzed using ECHO (v3.22) to determine the molecular clock period (τ) of rhythmic strains, using the smoothing, normalizing, and linear detrending options[41]. Final plots of moving average data and standard deviation were made in PRISM 9.0.2, for both original and post-ECHO normalized/detrended data. For original camera trials and extracted values, see [https://doi.org/10.17632/7hgspb5gn7.1].

## Co-immunoprecipitation and western blotting

Tissue from each *N. crassa* strain was grown in triplicate, by making a conidial suspension using 1 mL of Liquid Culture Media (LCM; 2% Glucose, 0.5% Arginine, 1× Vogel's Salts, 50 ng/mL Biotin) to resuspend conidia from a ~1 week old Vogel's minimal media slant. The centrifuged and washed conidia were then inoculated into a 125 mL flask containing 50 mL of LCM and grown at 25 °C, 125 rpm shaking in constant light for 48 h before harvesting. Harvesting was carried out using vacuum filtration and flash freezing in Liquid Nitrogen before storing at −80 °C. Tissue was ground in a mortar and pestle along with Liquid Nitrogen, before extracting tissue lysate using a similar volume of chilled Protein Extraction Buffer was added (pH 7.4, 50 mM HEPES, 137 mM NaCl, 10% Glycerol, 0.4% NP-40 alternative (Calbiochem, 492016)) with 1× HALT protease and phosphatase inhibitor, EDTA-free (ThermoScientific, 87785). Co-Immunoprecipitation employed 40 µL of resuspended anti-FLAG (M2 clone, mouse) magnetic beads (M8823, Sigma) or anti-V5 (clone V5010, mouse) agarose beads (A7345, Sigma) that were prepared according to manufacturer's instructions for each 3.5–4.8 mg of total lysate, as measured by Bradford assay. Final volumes were brought up to 1 mL total with further Protein Extraction Buffer with 1× HALT. Samples were incubated overnight at 4 °C while nutating. Following placement on a magnetic rack, the flow through was removed and the beads were washed three times with 1 mL of Protein Extraction Buffer. Eluted proteins were retrieved by adding 40 µL of 2× LDS Buffer (Life Technologies, NP0008) and boiling for 15 min. at 65 °C. Samples were removed to a fresh Eppendorf tube and boiled at 100 °C for 5 min. with the addition of 3%v/v ß-mercaptoethanol before freezing at −20 °C. Thawed samples were run on precast NuPAGE 3–8% Tris-Acetate gels (Invitrogen, WG1602BOX) following manufacturer's protocol, and transferred to PVDF membrane using a Bio-Rad Trans-Blot Turbo Transfer System (Bio-Rad, 1704150). Membrane was blocked using 5% Milk in PBS buffer with 0.2% Tween-20, Primary antibodies that were custom polyclonal antibodies produced in rabbit, were courtesy of the Dunlap-Loros Labs at Dartmouth, and were used at 1:5000 for anti-FRQ[103], 1:12,000 for anti-FRH[71] or 1:5000 for anti-WC-1[104], in 1% Milk with PBS and 0.2% Tween-20. The secondary antibody was 1:5000 Goat anti-Rabbit polyclonal (Invitrogen, 31460) in PBS and 0.2% Tween-20. Western blots were incubated with SuperSignal West FEMTO (Thermo-Scientific, 34094) or ATTO (ThermoScientific, A38554) for anti-WC-1

blots, and imaged using a Bio-Rad GelDoc imager. The commercial primary antibody used to detect short-ck1a[HA] was anti-HA (clone 2-2.2.14, mouse) at 1:10,000 dilution (Invitrogen, 26183), along with Goat anti-Mouse polyclonal secondary at 1:10,000 (Invitrogen, 31430). Anti-HA western blots were incubated with SuperSignal West PICO (ThermoScientific, 34577) and imaged as above. Image Lab software (version 6.0.1) was used for the relative quantification of elution bands for FRQ versus FRH and CK1a. As this was a comparison across blots, it is important to note that these blots were processed in parallel for each of the 3 biological replicates. Due to different distributions, a non-parametric Mann-Whitney U-test was used within PRISM 9.0.2. See the Source Data file for full-size blots with layered colorimetric ladders and associated amido-stained membranes.

### Cycloheximide assay and semi-quantitative western blotting

The cycloheximide assay was adapted from Hurley et al.[19]. Briefly, conidial suspensions from the designated *N. crassa* strains were inoculated into a petri dish filled with LCM (2% glucose) and grown in constant light at 25 °C. After 24–36 h (dependent upon the growth rate of the strain) plugs were cut from the resulting mycelial mat and placed into individual Erlenmeyer flasks with -50 mL of LCM (2% glucose). After plugs grew in constant light at 25 °C (125 rpm shaker) for a further 24 h, 40 μg/uL of Cycloheximide (94271, VWR) was added to each culture and this was designated time 0. Samples continued to grow at LL, 25 °C, 125 rpm shaker until they were harvested using vacuum filtration at different timepoints (0, 2, 3, 4, 5, 6, 7 h post-cycloheximide addition) via flash freezing in liquid nitrogen before storing at −80 °C. Protein extraction and standardization were executed as described above. SDS-PAGE and Western blot protocols were carried out as above, except using precast NuPAGE 4–12% Bis-Tris gels (Invitrogen, WG1402BOX), 1:5000 anti-V5 (clone SV5-Pk1, mouse) primary (Invitrogen, 46-1157) and 1:25,000 Goat anti-Mouse polyclonal (Invitrogen, 313430) secondary antibodies, and SuperSignal West FEMTO (ThermoScientific, 34094) or ATTO (ThermoScientific, A38554) in the case of FRQ[FFD2]. Image Lab software (version 6.0.1) was used for relative quantification of bands and Amido Black staining was used as a total protein loading control for normalization. Pixel density was quantified from hour 0–7 or 0–3 h within each western blot (dependent upon the stability of FRQ) and normalized by dividing by the pixel density of a matching area of the Amido Black stained membrane. Normalized data was plotted as a ratio relative to time-point 0 in PRISM 9.0.2, and this program was used to carry out the two-way ANOVA and Tukey's post hoc tests. For the WC-1 relative lysate levels, pixel density after dividing by the total protein measured from Amido Black stained membrane, was plotted as a ratio relative to FRQ[VHF] WC-1 levels, and PRISM 9.0.2 was again used to carry out a one-way ANOVA and Tukey's post hoc tests. See the Source Data file for full-size blots with layered colorimetric ladders and associated amido-stained membranes.

### AlphaFold2 structural modeling

The modeled 50 a.a. portion of FRQ (centered on the FFD region) used as the basis of the FRQ images presented in Fig. 6e was done using a Google Colab notebook based on a simplified version of AlphaFold v2.1.0[105], made available at the following website: https://colab.research.google.com/github/deepmind/alphafold/blob/main/notebooks/AlphaFold.ipynb.

### Statistics and reproducibility

In addition to the sample sizes, exclusion criteria, and statistics discussed in the rest of the Methods section, please also refer to the Reporting Summary for more details on each of these topics. Please note that no sample sizes were predetermined and our study did not involve treatment groups necessitating randomization or blinding.

### Reporting summary

Further information on research design is available in the Nature Portfolio Reporting Summary linked to this article.

## Data availability

The authors declare that the data supporting the findings of this study are available within the paper and its supplementary materials. The designed FRQ peptide libraries, original FRQ peptide microarray images, extracted peptide microarray intensity values, luciferase reporter CCD array trials, and relative luminescence values are included in Mendeley Datasets (https://doi.org/10.17632/7hgspb5gn7.1). All associated inputs, subsampled trajectories, and analyses related to the All-Atom Monte Carlo simulations are available on Zenodo (https://zenodo.org/records/10793684). Additionally, a previously solved crystal structure of FRH that appears in Figs. 2d and 6e can be found in the PDB under accession code 5E02. Source data are provided with this paper.

## Code availability

Code used in the bioinformatic analysis, simulation analysis, subsampled trajectories information, and simulation input information can be found at Zenodo (https://zenodo.org/records/10793684).

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

## Acknowledgements

We thank the Fungal Genetic Stock Center and the Dunlap and Loros labs at Dartmouth College for providing some *Neurospora crassa* strains. Lab technical support was kindly provided by Christopher Kirchhoff, Ellinor Tai, Samantha Keller, and Sandhya Vellayappan. We thank Sergey Pryshchep and the RPI Core for the use of equipment. This work was supported by an NIH-National Institute of General Medical Sciences T32 Fellowship GM067545 (to M.S.J.), an NSF Graduate Research Fellowship DGE-2139839 (to D.G.), an NSF Graduate Research Fellowship DGE-1247271 (to D.G.S.), a Longer Life Foundation collaboration between RGA and Washington University (to A.S.H.), a Human Frontiers in Science Program (HFSP) Research Grant RGP0015/2022 (to A.S.H.), an NIH-National Institute of Biomedical Imaging and Bioengineering Grant U01EB022546 (to J.M.H.), an NIH-National Institute of General Medical Sciences Grant R35GM128687 (to J.M.H.), an NSF CAREER Award 2045674 (to J.M.H.), and Rensselaer Polytechnic Startup funds (to J.M.H.).

## Author contributions

Conceptualization, J.M.H.; Methodology, M.S.J., D.G., D.G.S., J.F.P., P.K., A.S.H., J.M.H.; Investigation, M.S.J., D.G.S., J.F.P., J.T.; Formal analysis, M.S.J., D.G., D.G.S, J.F.P, G.M.G., J.T.; Software, D.G., G.M.G., A.S.H.; Resources, P.K., A.S.H., J.M.H.; Writing – original draft, M.S.J., D.G., J.F.P.; Writing – review & editing, M.S.J., D.G., D.G.S., J.F.P., A.S.H., J.M.H.; Visualization, M.S.J., D.G., D.G.S., J.F.P.; Supervision, M.S.J., P.K., A.S.H., J.M.H.; Funding acquisition, M.S.J., D.G., D.G.S., A.S.H., J.M.H.

## Competing interests

The authors declare no competing interests.
