## [Peer Review File · Nature Communications]

Disordered clock protein interactions and charge blocks turn an hourglass into a persistent circadian oscillatorEditorial Note: This manuscript has been previously reviewed at another journal that is not operating a transparent peer review scheme. This document only contains reviewer comments and rebuttal letters for versions considered at *Nature Communications*.

Reviewer #1 (Remarks to the Author):

This revised manuscript has improved clarity and I thank the authors for taking the time to address my earlier comments. I think this is a strong contribution towards understanding the function of IDRs/SLiMs in clock proteins and recommend it be accepted for publication.

Reviewer #2 (Remarks to the Author):

The focus of my review is to evaluate whether authors addressed concerns raised by reviewers in the previous round of revisions. In their revision, the authors have prioritized the point-by-point response without committing to make changes in the main text that directly address each point raised by reviewers. As a result, several major issues in the manuscript persist. Below I highlight several areas of the manuscript that remain sticky points and would benefit from improvement:

1) Unaddressed major concern #1: All reviewers raised various concerns about the specificity of hits coming from the LOCATE assay. These concerns are justified (and were explained in detail in each of the provided reviews), but they are not well addressed in the authors' response. Given that the authors base many claims on this assay, it is critical that they address this particular concern with more rigor. In addition to other reviewers' previous concerns, it is also not clear to me how the authors controlled for non-specific binding in their LOCATE assay. For this assay, authors used His-tagged FRH (where the desired construct represents only ~30% of purified protein content) as prey, which is also affinity-tagged. Did the authors counter-screen with an empty His-tag prepared in a similar way to identify non-specific binders? If so, this should be shown. Given the major dependency of the manuscript on the LOCATE data, if the authors did not do this, at a minimum, this needs to be acknowledged as a limitation of the study design.

2) Unaddressed major concern #2: As pointed out by multiple reviewers, there is no further evidence beyond the LOCATE findings that the charged blocks' role on the molecular clock is through FRH. Concerningly, in contrast to the LOCATE data, the mutation of this region did not affect the FRH:FRQ interaction in cells, so authors conclude that it is likely regulated through non-specific, charged interactions. Without properly addressing concern #1 (whether the LOCATE hits might be false positives), this aspect of the paper is very hypothetical and may possibly be wrong. As a result, the central conclusion that the FRH:FRQ interaction is mediated by "fuzzy" interactions is currently not well justified nor supported by the data. I would strongly suggest that the specific claims about the nature of the biochemical interaction that depend solely upon the LOCATE assay be removed or at a minimum not be relied upon for the central message.

3) Unaddressed major concern #3: As multiple reviewers previously pointed out, large parts of the paper are hypothetical/speculative, in the results section in particular. Authors should make a more focused effort to write the results section to focus strictly on conclusions they can make given their experimental data.

4) Unaddressed concern #4: Response to the concern about novelty. Authors argue in their response to reviewer #2 that the newly identified SLiM region behaves differently than the previously identified SLiM and suggest this behavior to be a novel unexpected role in the molecular clock (FRH helps the clock to persist and regulate output). Unfortunately, the central message of the manuscript does not prioritize this discovery, but instead emphasizes IDR-related conclusions that are weakly supported, such as the biochemical basis of interactions that are inconsistent in vitro and in vivo.

5) Unaddressed concern #5: Related to the previous concern, authors state in their response that the major novelty/impact is on the molecular clock field and they tried to frame the paper in light of that. However, the manuscript frequently employs language from the fields of IDRs and structural biology, which are not the major disciplines likely to be impacted by the work. Simpler, more strategic usage of technical IDR-focused language (especially in the results section) may increase the overall impact on the molecular clock field.

Reviewer #3 (Remarks to the Author):

The continued absence of an orthogonal method to validate direct binding of any of these FRQ peptides to FRH, and the failure to provide any support for their claim of the multivalency of peptide binding apart from the microarray results, are serious flaws in this study. The microarray data, even if taken at face value, only suggest the possibility of multivalent binding of these charge islands/blocks, but in no way do they provide definitive proof of it. A fluorescence polarization assay looking at FRH binding to >1 peptide simultaneously, each labeled with a different compatible fluorophore, would provide much-needed independent biochemical support for their claims. It is not necessary to validate all of the peptide results, simply to provide results from an orthogonal binding assay (preferably under equilibrium conditions) that would give unambiguous evidence of direct and multivalent interactions. The authors even state in the rebuttal that "By themselves, they [the microarray results] do not show that the complex is multivalent. However, we disagree with the reviewer, as described in more detail above, and believe that this does show that there is evidence for charge driven binding between FRH and FRQ peptides." It is not clear what they are referring to (above), and in any regard, no one is contesting the likely charge-driven nature of the interactions, just that there is insufficient data to support other major claims and the title of the manuscript.

Moreover, the repeated claims in the rebuttal that biochemical or biophysical validation of the central premise would take years are an exaggeration. Certainly, chasing down the structural and functional implications of each SLiM in a protein this size could take a whole career--but surely, one should rigorously validate the central premise of the study? While the in vivo data suggest that the authors are likely on an interesting trajectory with this work, it does not provide support for their claims of direct and multivalent interactions. If you want to make a biochemical claim, you should provide rigorous support for it, especially for a journal of this stature. With synthesized peptides and FRH protein in hand, FP assays for a set of peptides (and mutants) could be completed in a few weeks. However, this would require working with pure FRH protein, something the authors have apparently been unwilling or unable to do. While it is understood that peptide microarrays can be run with impure protein or even lysate, these conditions introduce uncertainty in the interpretation of the results (is the binding direct or not?), diminishing the rigor of the work in the absence of other confirmatory assays. It is also understood that, as a tiled peptide microarray, this is not a new method, which begs the question of why the authors bothered to give it a clever name (i.e., LOCATE)? The citation listed for the assay on line 98 has nothing to do with LOCATE as far as I could tell.

REVIEWER COMMENTS

Before we get into the point-by-point response, we would like to take the time to thank both the reviewers and editor for their fantastic suggestions related to how to improve this manuscript. We feel that the suggested revisions have helped us to craft this manuscript so that it is accessible and relevant to the fields of research that we hope to contribute to. In this revision, we strive to ensure that all our conclusions are supported and, when we speculate on the meaning of these conclusions, that speculation is clear to the reviewer and future reader. Where we have made changes to respond to this need are underlined in the response to reviewers and highlighted in the main text.

Reviewer #1 (Remarks to the Author):

This revised manuscript has improved clarity and I thank the authors for taking the time to address my earlier comments. I think this is a strong contribution towards understanding the function of IDRs/SLiMs in clock proteins and recommend it be accepted for publication.

-We thank you for your time in providing comments that greatly contributed to our manuscript. We are excited to share this work with both the clock and disordered protein fields.

Reviewer #2 (Remarks to the Author):

The focus of my review is to evaluate whether authors addressed concerns raised by reviewers in the previous round of revisions. In their revision, the authors have prioritized the point-by-point response without committing to make changes in the main text that directly address each point raised by reviewers. As a result, several major issues in the manuscript persist. Below I highlight several areas of the manuscript that remain sticky points and would benefit from improvement:

-We appreciate and thank you for the time and effort that you have put into the reviews and will continue to work to clarify any points that you feel are still limitations of the paper.

1) Unaddressed major concern #1: All reviewers raised various concerns about the specificity of hits coming from the LOCATE assay. These concerns are justified (and were explained in detail in each of the provided reviews), but they are not well addressed in the authors' response. Given that the authors base many claims on this assay, it is critical that they address this particular concern with more rigor. In addition to other reviewers' previous concerns, it is also not clear to me how the authors controlled for non-specific binding in their LOCATE assay. For this assay, authors used His-tagged FRH (where the desired construct represents only ~30% of purified protein content) as prey, which is also affinity-tagged. Did the authors counter-screen with an empty His-tag prepared in a similar way to identify non-specific binders? If so, this should be shown. Given the major dependency of the manuscript on the LOCATE data, if the authors did not do this, at a minimum, this needs to be acknowledged as a limitation of the study design.

-We agree with the reviewer that the LOCATE assay is an important part of our manuscript and want both the reviewers and readers to believe that these results are rigorous. For this reason, we collaborated with our co-authors, Dr. Shastry and Dr. Karande, who are experts in the field of peptide microarrays and have spent many years learning the proper and exacting controls necessary to garner relevant data from microarrays (e.g. ¹). We feel that it is important to note here that peptide microarrays as a technology are not new. Rather, this approach has been

extensively tested to show that the results of a peptide array reflect binding relationships *in vivo*. In this manuscript, we refocused this established technology on a new application, the problem of how to identify potential binding regions within IDRs, especially highly disordered clock proteins like FRQ in *Neurospora*. Given their knowledge base, we and our collaborators implemented a peptide microarray experimental strategy using stringent blocking procedures (with BSA), washes, a variety of concentrations of FRH, and more than one antibody to visualize results. This approach allowed us to ensure that the pattern of FRQ peptide binding FRH was repeatable across peptide arrays and protein concentrations, the results were not antibody-specific, and the interaction reflected the actual binding of FRH protein to FRQ peptides. Beyond the tiling of peptides from an original protein sequence, LOCATE included the rational design of peptides (including scrambles, truncations, and point mutations of potential binding regions), some of which appear by necessity in our extended data figures. These rationally designed peptides demonstrated that the binding was not spurious and allowed us to pinpoint the important residues for binding, such as the double arginines in the extended FFD SLiM.

-Having noted the above, we appreciate the concerns of the reviewer and so would like to summarize these concerns as we understood them, and our responses during this and previous review processes, in the hope of mitigating the reviewer's concerns.

- 1) In the first round of reviews Reviewer #1 (Main point #1) and Reviewer #3 (Major #4) had concerns that we did not show FRH binding specificity for the additional binding regions outside of the FFD-region peptides. We believed these concerns were not related to the lack of rigorous controls but to our lack of clarity in distinguishing the difference between the sequence specific interaction of the FFD SLiM and the electrostatic interaction between some of the other FRQ peptides and FRH. As a review, the LOCATE assay showed clear evidence of sequence-specific binding within the FFD SLiM, a region that has consistently been shown by us and others to be essential for the formation of a complex between FRQ and FRH ^{2,3}. Contrary to this, LOCATE also showed additional regions of FRH binding to FRQ-peptides that were not sequence specific, but had an overall positive charge, which was not surprising given our additional evidence of the negative electrostatic surface potential of FRH (which Reviewer #1 agreed with in Main Point #1). We found that these regions of positive charge were not distributed randomly within FRQ but exist in significant clusters or "blocks" along the primary sequence of FRQ. This prompted us to follow-up as to why FRQ might have these charge blocks (which we showed to be a shared feature with the functional PER homologs) and what effect mutating a positive charge block might have on clock function. To clarify these points, we made textual changes to define these two modes of interaction (sequence-specific SLiM versus electrostatics) throughout the intro, results, and discussion in revision #1. In response to Reviewer #2's request in revision #1 (last Primary issue) for a conceptual schematic to integrate how these molecular interactions relate to how the (circadian) system works, we developed Summary Fig. 6e. This very helpful request increased the clarity of our working model, and we believe will help our future readers understand one of the main insights of our manuscript: that there are multiple modes of interaction between FRH and FRQ. While a sequence-specific SLiM is necessary for complex formation, electrostatic interactions are also involved in the FRQ:FRH complex.
- 2) Additionally in the first round of reviews, Reviewer #3 thought we had used degraded FRH protein (Major #1). If this was the case, our LOCATE results would certainly be invalid. To alleviate this concern, we provided the western blot that was done at the time of our FRH expression, showing that our anti-His antibody only detected full-length FRH (Ext. Data. Fig. 1c) and there were no smaller degradation products of FRH, demonstrating that the non-specific controls in our FRH sample were what we had stated they were, non-specific

bands from the lysate (Ext. Data Fig. 1b). Our co-authors ensured that we met the field's standard by having us use two different antibodies; anti-FRH, raised against only FRH a.a.'s 1 to 374 as noted by Reviewer #3 (in Round 2, "Other comments") and anti-His to detect the epitope tag on the N-terminal end of the protein. If, beyond these controls, there were still somehow degradation products of FRH present, there would be a low likelihood that the microarray results were highly correlated between the different FRH concentrations and different antibodies used. To clarify this point, we have added a more explicit statement regarding this in line 139-140 of our newly revised manuscript. In revision #1, correlation plots were added to show that there was a high correlation of binding intensity on the microarray between replicates using different estimated concentrations of FRH, as well as a high correlation in the results obtained when using anti-His versus anti-FRH to visualize results (Ext. Data Figs. 1d and 1e), supporting that the interactions in the LOCATE assay are between FRQ and FRH.

- 3) To respond to the concern regarding specificity in this revision, we believe that the reviewer feels that the binding we see in our LOCATE assay could potentially be explained by the His epitope-tag itself that is present on FRH. Having spoken to our co-authors, this is an unlikely source of interaction and is not a typical field assessment used in peptide microarrays, thus they did not recommend we include this as one of our controls. However, we would like to ensure that this reviewer concern, which could be shared by other readers, is addressed. Therefore, we would like to highlight two important points. First, if binding was occurring due to the His-epitope tag, then there should be no evidence of sequence-specific binding within our FFD-related peptides (tiled "peptide map", or follow-up scrambles, truncations, and point mutations). However, there is clear specificity in this interaction (Extended Data Figure 4). Second, as histidine is slightly protonated at pH 7.0, it is very likely that the overall preference for the His-tag should be to negatively charged FRQ peptides rather than positively-charged FRQ peptides. Nevertheless, we agree that this is another control that could have been run and have added a caveat to our discussion (lines 478-481), that while we did not employ this potential control, the readers may need to consider this potential confounding effect when applying this method to their own candidate set of interactors.

2) Unaddressed major concern #2: As pointed out by multiple reviewers, there is no further evidence beyond the LOCATE findings that the charged blocks' role on the molecular clock is through FRH. Concerningly, in contrast to the LOCATE data, the mutation of this region did not affect the FRH:FRQ interaction in cells, so authors conclude that it is likely regulated through non-specific, charged interactions. Without properly addressing concern #1 (whether the LOCATE hits might be false positives), this aspect of the paper is very hypothetical and may possibly be wrong. As a result, the central conclusion that the FRH:FRQ interaction is mediated by "fuzzy" interactions is currently not well justified nor supported by the data. I would strongly suggest that the specific claims about the nature of the biochemical interaction that depend solely upon the LOCATE assay be removed or at a minimum not be relied upon for the central message.

-We are pleased that the reviewer agrees that the positive charge blocks play a role in the molecular clock. The shared molecular feature of positive charge blocks in repressive arm clock proteins, and the severe clock phenotype of the mutation of one of these blocks, are exciting examples of how IDR chemical features are important for circadian timing.

-Unaddressed major concern #2 appears to be interrelated with Unaddressed major concern #1. For the above-described reasons, we feel our LOCATE results are valid and can be used to reasonably drive the downstream research. Therefore, we will limit our response in this section to the concerns brought up that are specific to Unaddressed major concern #2. Based on this point,

it is clear the reviewer has continuing concerns about our interpretation of our results. Before going into a more detailed discussion, we wanted to re-iterate that we agree that our working model of a “fuzzy-like” complex is one of a few potential interpretations of our results. However, this model is in keeping with the results of other recent studies (eg.⁴⁻⁶). We have sought to make clear in our manuscript within the discussion that this is our proposed hypothesis/model that is intended to be useful for informing future studies and generating testable hypotheses but has not been fully proved. To support this distinction, we have changed our manuscript title to better distinguish the findings from the model.

- 1) In major concern #2, the reviewer suggests that the change in function based on mutation of the electrostatic regions may be due to changes in the interaction with a different protein. This supposition is related to a concern expressed by Reviewer #3 in the first round of reviews (Major #2), where they noted that the mutation was too close to the CK1A binding site (known as FCD-1). Reviewer #3 supposed that the clock effects may be due to a change in FRQ-CK1A binding. We therefore created a novel strain (CK1A:HA tag) to allow for Co-IPs of FRQ that would allow us to track FRQ’s interaction with CK1A. We did not detect a difference in CK1A binding with this experiment (new quantification added to this revision, see lines 260-263 and Extended Data Fig. 3e). We also added an important caveat to our discussion in response to this concern in this revision (lines 432-436), that “The LOCATE data in combination with the pull downs suggests this effect is due to a change in FRQ/FRH interaction at secondary sites. However, our assay cannot completely rule out that this is due to a change in FRQ interaction with other proteins, though the possibility that a change in interaction between FRQ and FRH leads to a change in the FRQ macromolecular complex is in keeping with our proposed model” However, given this data and the data from LOCATE, which we believe the evidence supports is a valid analysis of the FRQ:FRH interaction, the most likely explanation is that there is a change in the electrostatically driven interaction between FRQ and FRH.
- 2) Also in major concern #2, the reviewer reports they are troubled that the LOCATE interaction results are different in some cases from the *in vivo* interaction results, as is the case in the electrostatic interactions between FRQ and FRH. Notably, one of the reasons that we chose to use a peptide microarray was that it has been validated that peptide microarrays have an added benefit of capturing a wider range of interactions, including “weaker” interactions such as electrostatic interactions that cannot be found using pull down assays⁷. As part of a previous revision, we had added language to this effect within the manuscript, in lines 424-426. In fact, it is an integral part of our model that we were able to capture FRH interactions with positively charged peptides from regions of FRQ on the peptide microarray (and believe these results are valid based on our response to major concern #1 above). The fact that these interactions are not sufficient to allow a complex to form *in vivo* between FRQ and FRH is what led us to propose the fuzzy-like model. Moreover, the unusual clustering of charged amino acids to form charge blocks in FRQ (and other functional PER homologs), as well as the surface electrostatic potential of FRH, supports the LOCATE results. Perhaps most exciting is that while we did not lose the FRQ-FRH interaction *in vivo* by mutating the KKK315AAA FRQ charge block, we did see a change in regulation of circadian physiology. However, we want to highlight that our main conclusion from the data is that these positive charge blocks in FRQ have a circadian function, while in our discussion we bring these results together as part of our proposed model of a “fuzzy-like” complex. We understand this reviewer’s desire to have this message clarified and, in this revision, we have gone through the manuscripts to make this message clear, changing the below sections to clarify the message.

-Former lines 340-342, now in Discussion lines 422-424.

-Former lines 268-270, now in Discussion lines 430-432.

3) Unaddressed major concern #3: As multiple reviewers previously pointed out, large parts of the paper are hypothetical/speculative, in the results section in particular. Authors should make a more focused effort to write the results section to focus strictly on conclusions they can make given their experimental data.

-Thank you for this point. In past revisions, our efforts to move all speculation to the discussion based on the reviewer's previous comments have improved the flow of the manuscript and made it clear what are results and what was informed interpretation and speculation that led to our model of the potential fuzzy-like complex of FRQ and FRH (Fig. 6). After a careful re-reading of this version of the manuscript, we found a few remaining statements that linked results to our later interpretations in the discussion. To minimize these points of speculation, we have removed any elements of speculation and now integrate the below points only in the discussion section:

-Former lines 227-230, now in Discussion lines 451-454.

-Former lines 361-365, now in Discussion lines 395-396.

-Former lines 392-400, now in Discussion lines 413-419.

-As our proposed model is still speculative but is a working model leading to important future studies, we have added our next steps to the Discussion, again in lines 474-478.

4) Unaddressed concern #4: Response to the concern about novelty. Authors argue in their response to reviewer #2 that the newly identified SLiM region behaves differently than the previously identified SLiM and suggest this behavior to be a novel unexpected role in the molecular clock (FRH helps the clock to persist and regulate output). Unfortunately, the central message of the manuscript does not prioritize this discovery, but instead emphasizes IDR-related conclusions that are weakly supported, such as the biochemical basis of interactions that are inconsistent in vitro and in vivo.

-We appreciate that the reviewer, not being in the circadian field, recognized the importance of our new findings related to which proteins are playing a role in circadian repression versus persistence. This is a very exciting distinction. At the suggestion of the reviewer, we had modified our title to highlight this more readily in past revisions. However, we feel that our work at the interface of the IDP and clocks field is as exciting and wish to continue to highlight this area as well. The understanding of how clock proteins may have a conserved mechanism across phyla that allow them to work together to keep time and control physiology through multi-valent interactions has very exciting implications. Our proposed model could explain the mechanism of how phosphorylation alters clock proteins to enable their functionality. While there is certainly much more to do to evaluate this working model, without placing our findings within the working model we proposed, we feel it wouldn't be as useful to the community.

5) Unaddressed concern #5: Related to the previous concern, authors state in their response that the major novelty/impact is on the molecular clock field and they tried to frame the paper in light of that. However, the manuscript frequently employs language from the fields of IDRs and structural biology, which are not the major disciplines likely to be impacted by the work. Simpler, more strategic usage of technical IDR-focused language (especially in the results section) may increase the overall impact on the molecular clock field.

-We thank the reviewer for their thoughts on how we can modify the manuscript to make it more accessible to the field of circadian rhythms. It is important for the clock field to understand disordered proteins as there is a great deal of protein disorder in the components of the molecular clock. While the explanation of what we are hypothesizing necessitates the use of IDR-focused language, we could have better defined this language within the manuscript. We have tackled this in previous reviews, including the change from charge “islands” to charge “blocks” as defined in ⁸. Further, this reviewer’s previous suggestion to broaden the impact of our manuscript by providing insights on the implications of our work beyond the circadian field (Round 1, “Other Issues”, first paragraph) was an excellent suggestion that led to a stronger ending of our discussion that makes the manuscript more inter-disciplinary. Considering the further concern of this reviewer, we have endeavored to make the language more accessible to those in the circadian field. To that end, we have simplified the language in our discussion to make it more approachable to a broader range of readers. This, along with the change in the introduction at this reviewer’s request, creates a more accessible and approachable manuscript for the circadian field.

Reviewer #3 (Remarks to the Author):

The continued absence of an orthogonal method to validate direct binding of any of these FRQ peptides to FRH, and the failure to provide any support for their claim of the multivalency of peptide binding apart from the microarray results, are serious flaws in this study. The microarray data, even if taken at face value, only suggest the possibility of multivalent binding of these charge islands/blocks, but in no way do they provide definitive proof of it. A fluorescence polarization assay looking at FRH binding to >1 peptide simultaneously, each labeled with a different compatible fluorophore, would provide much-needed independent biochemical support for their claims. It is not necessary to validate all of the peptide results, simply to provide results from an orthogonal binding assay (preferably under equilibrium conditions) that would give unambiguous evidence of direct and multivalent interactions. The authors even state in the rebuttal that “By themselves, they [the microarray results] do not show that the complex is multivalent. However, we disagree with the reviewer, as described in more detail above, and believe that this does show that there is evidence for charge driven binding between FRH and FRQ peptides.” It is not clear what they are referring to (above), and in any regard, no one is contesting the likely charge-driven nature of the interactions, just that there is insufficient data to support other major claims and the title of the manuscript.

Moreover, the repeated claims in the rebuttal that biochemical or biophysical validation of the central premise would take years are an exaggeration. Certainly, chasing down the structural and functional implications of each SLiM in a protein this size could take a whole career—but surely, one should rigorously validate the central premise of the study? While the in vivo data suggest that the authors are likely on an interesting trajectory with this work, it does not provide support for their claims of direct and multivalent interactions. If you want to make a biochemical claim, you should provide rigorous support for it, especially for a journal of this stature. With synthesized peptides and FRH protein in hand, FP assays for a set of peptides (and mutants) could be completed in a few weeks. However, this would require working with pure FRH protein, something the authors have apparently been unwilling or unable to do. While it is understood that peptide microarrays can be run with impure protein or even lysate, these conditions introduce uncertainty in the interpretation of the results (is the binding direct or not?), diminishing the rigor of the work in the absence of other confirmatory assays. It is also understood that, as a tiled peptide microarray, this is not a new method, which begs the question of why the authors bothered to give

it a clever name (i.e., LOCATE)? The citation listed for the assay on line 98 has nothing to do with LOCATE as far as I could tell.

-We are pleased to hear that the reviewer agrees that electrostatics are likely a major player in the interaction between FRQ and FRH. We also understand the reviewer (like us!) is excited for further experiments to continue to investigate the findings in this manuscript. Indeed, further *in vitro* orthogonal methods are a long-term goal for our research. Regardless, we believe these are outside the scope of the current study. To explain our position as clearly as possible, our initial motivation in this study was to investigate ways in which we can quickly identify candidate IDRs of interest, tested with follow-up scrambles, point mutations, and truncations, all within one accessible assay. Having adapted protein microarrays to our need, we employed LOCATE and discovered unanticipated findings related to the interaction between FRQ and FRH. One of these was that our assay identified new critical residues that were important for the FFD SLiM, which we then verified *in vivo*. This allowed us to uncover that FRH is not actually needed for repression but supports clock persistence. Further, we discovered interaction regions that involve positive charge blocks within FRQ and that these blocks are a consistent feature in higher eukaryotes. Mutating one of these charge blocks affected clock regulation of physiology without breaking the feedback loop that allows the clock to keep time. Bringing these results together, we proposed a working model within the discussion of a fuzzy-like complex. The validity of the nature of this complex will be the focus of future work, but it is the usefulness of the LOCATE assay itself and the insights it has already yielded that we wish to share in this manuscript.

-To parse the request for orthogonal experiments further, the reviewer suggest that we perform fluorescence polarization to validate this interaction. While fluorescence polarization has many strengths, we do not believe it would be an appropriate application for the analysis of our complex. First, the characterization of individual Kd's may not fully describe the complex contributions of electrostatics to the FRQ:FRH interaction. Rather, we'd like to investigate the influence of these multi-valent interactions using SPR, cryo-EM, and SAXS to describe the structure of the complex. We have updated our discussion section to note these future goals. This will undoubtedly take much trial and error and many years to complete and we do not want to hold up the exciting findings described above while we work to create this more detailed model of FRQ:FRH interaction.

-The reviewer accurately points out that tiled peptide microarrays are indeed not a new method. In fact, this as a strength for our work as it meant that the microarrays do not require a full validation and the standards for rigorous controls have already been established. It is relevant to this discussion that peptide microarrays showing multiple binding regions that were validated *in vivo* have been previously published (e.g. ⁹). However, our assay is a new application that uses the rational design of peptides to understand the overall binding behavior and find the SLiMs of an interactor for a largely disordered protein. Finally, the citation listed on line 98 relates to a paper by our co-authors ¹, which employs rational design and peptide microarrays, but is indeed not the LOCATE assay which is instead presented in this paper.

References

1. Shastry, D. G. & Karande, P. Microarrays for the screening and identification of carbohydrate-binding peptides. *Analyst* **144**, 7378–7389 (2019).
2. Cheng, P., He, Q., He, Q., Wang, L. & Liu, Y. Regulation of the *Neurospora* circadian clock by an RNA helicase. *Genes Dev.* 234–241 (2005) doi:10.1101/gad.1266805.
3. Guo, J., Cheng, P. & Liu, Y. Functional Significance of FRH in Regulating the Phosphorylation and Stability of *Neurospora* Circadian Clock Protein FRQ. *J. Biol. Chem.* **285**, 11508–11515 (2010).
4. Pelham, J. F., Mosier, A. E. & Hurley, J. M. Characterizing Time-of-Day Conformational Changes in the Intrinsically Disordered Proteins of the Circadian Clock. *Methods. Enzymol.* **611**, 503–529 (2018).
5. Tariq, D. *et al.* Phosphorylation, disorder, and phase separation govern the behavior of Frequency in the fungal circadian clock. *bioRxiv* 2022.11.03.515097 (2022) doi:10.1101/2022.11.03.515097.
6. Wang, B. & Dunlap, J. C. Domains required for the interaction of the central negative element FRQ with its transcriptional activator WCC within the core circadian clock of *Neurospora*. *J. Biol. Chem.* **299**, 104850 (2023).
7. Katz, C. *et al.* Studying protein-protein interactions using peptide arrays. *Chem. Soc. Rev.* **40**, 2131–2145 (2011).
8. Lyons, H. *et al.* Functional partitioning of transcriptional regulators by patterned charge blocks. *Cell* **186**, 327–345 (2023).
9. Dittmar, G. *et al.* PRISMA: Protein Interaction Screen on Peptide Matrix Reveals Interaction Footprints and Modifications-Dependent Interactome of Intrinsically Disordered C/EBP β . *iScience* **13**, 351–370 (2019).
10. Volpato, A. *et al.* Extending fluorescence anisotropy to large complexes using reversibly switchable proteins. *Nature Biotechnology* **41**, 552–559 (2023).

11. Shi, M., Collett, M., Loros, J. J. & Dunlap, J. C. FRQ-interacting RNA helicase mediates negative and positive feedback in the *Neurospora* circadian clock. *Genetics* **184**, 351–361 (2010).
12. Conrad, K. S. *et al.* Structure of the frequency - interacting RNA helicase : a protein interaction hub for the circadian clock. *EMBO J.* **35**, 4–7 (2016).
13. Baker, C. L., Kettenbach, A. N., Loros, J. J., Gerber, S. A. & Dunlap, J. C. Quantitative Proteomics Reveals a Dynamic Interactome and Phase-Specific Phosphorylation in the *Neurospora* Circadian Clock. *Molec. Cell* **34**, 354–363 (2009).

Reviewer #1 (Remarks to the Author):

There is some good work here, it's not perfect but what paper is? The authors have used existing techniques in a new context to address an interesting and potentially important question. They provide plausible interpretations and caveats to for their findings and set out directions for future work. I strongly recommend accepting this paper for publication.

Reviewer #2 (Remarks to the Author):

Reviewers' comments:

Even though the authors made a significant effort to justify their interpretation of the data in the point-by-point response, the revision of the actual manuscript components did not address many concerns, as the authors made few substantial edits the manuscript text itself. Below I revisit the concerns that I previously raised.

Previous concerns #1 and #2:

I would like to thank the authors for adding the statement about possible false positive interactions due to the presence of the His-tag in the discussion. However, the second part of my previous concern remains unaddressed, described below:

In particular, statements related to FRQ:FRH interaction being driven by charged/electrostatic interactions are not supported by the data. (1) Mutating the charged residues in vivo did not disrupt FRQ:FRH interaction, and (2) an enrichment of charge in the LOCATE hits and/or identification of the existence of charged blocks is not the same as experimentally demonstrating that the charge of residues within these newly identified interfaces is important for the FRQ:FRH interaction. Authors agree that the additional validation of the importance of these charged surfaces is justified but will be done in future studies. I see no issue with authors putting forward their best interpretation of their data (Figure 6e) in the Discussion, especially since they make it clear that other interpretations of their data are possible as well. However, as a result of the above limitations of their experiments, more careful language should be used in the abstract, introduction, and results sections. The authors should not both state that these validations will be performed in the future, and also claim them as discoveries in this manuscript.

Statements like "clusters of positively charged residues within FRQ contribute to non-specific binding between FRQ and FRH," "While most FRH interaction with FRQ peptides in the LOCATE assay was driven by electrostatics," "Our method also revealed FRH can bind clusters of positively charged residues found within FRQ" or "FRH can bind clusters of positively charged residues found within FRQ" are not supported by the provided experimental data. The authors themselves point out in the Results section that it is possible that mutations of lysines to alanines could modify the conformational ensemble of FRQ in a way that disrupts molecular interactions rather than being charge-driven, therefore their conclusions about the role of charge in the text need to be more carefully stated.

For the same reason, statements about this interaction being driven by a Fuzzy complex in the abstract and last paragraph of the introduction are problematic. It is the authors' hypothesis in light of inconsistencies in vitro and in vivo, but it is not an experimentally supported conclusion.

Previous concern #3:

Portions of the Results section are still motivated by hypothetical conclusions. The most problematic is chapter #2 (FRH interacts predominantly with positively charged FRQ peptides, uncovering significant clusters of positive residues within FRQ), which is motivated by the assumption that the charge is the driving force behind the FRQ:FRH interaction in LOCATE because it is enriched in the peptides that were identified as hits, although not validated. The most obvious example (but not the only one), is in the last paragraph of this section, where the authors move ahead with their interpretation despite the lack of experimental support: "Given the hypothesized charge clusters and blocks, we next wondered if these could be dynamically modulated." Because the authors never experimentally establish their molecular interpretation, the text is not

scientifically sound. Authors should revise this section carefully and find a more accurate framing of their experimental results. I understand that some of these issues arose while authors tried to address reviewers concerns, however, it is authors' responsibility to work with the limitations of their data while ensuring readability and scientific soundness of the manuscript.

Previous concern #4:

My concern is not related to the presence of the final model in the discussion. My concern is related to the overall message of the abstract. Authors emphasize many IDR-related conclusions that are weakly supported and hypothetical, such as the biochemical basis of interactions, which are inconsistent in vitro and in vivo, rather than highlighting the novel unexpected roles in the molecular clock. Since authors argue in their previous response to reviewer #2 that the central novelty of their manuscript is the unexpected role of the described SLiMs in the molecular clock, the abstract should reflect that. As a result, this concern has not been addressed.

Previous concern #5:

This is stylistic, and might be a minor concern. Consistent with my previous review, large parts of the manuscript devoted to IDR-related analyses in the Results section remain very long, but lack in impact. The most problematic is the Results section corresponding to Figure 3a-c, which takes up more than 2 pages of text (lines 153-212). This reduces overall readability. A more concise and focused message in this section in particular may increase the overall impact on the molecular clock field.

RESPONSE TO REVIEWERS

-We would like to thank the editors and reviewers for their continued comments on this manuscript. We have made substantial changes in this round of revisions and look forward to the reviewers' further consideration.

Reviewer #1 (Remarks to the Author):

There is some good work here, it's not perfect but what paper is? The authors have used existing techniques in a new context to address an interesting and potentially important question. They provide plausible interpretations and caveats to for their findings and set out directions for future work. I strongly recommend accepting this paper for publication.

-We thank reviewer #1 again for taking the time to review our updated manuscript and are pleased that they still find our work interesting and recommend publication.

Reviewer #2 (Remarks to the Author):

Reviewers' comments:

Even though the authors made a significant effort to justify their interpretation of the data in the point-by-point response, the revision of the actual manuscript components did not address many concerns, as the authors made few substantial edits the manuscript text itself. Below I revisit the concerns that I previously raised.

-We are grateful to reviewer #2 for continuing to give us feedback on our manuscript. In this review, the reviewer was very specific about the changes that they wanted to see going forward, which was helpful in deciding how best to change the text to address the reviewers' concerns.

Previous concerns #1 and #2:

I would like to thank the authors for adding the statement about possible false positive interactions due to the presence of the His-tag in the discussion. However, the second part of my previous concern remains unaddressed, described below:

In particular, statements related to FRQ:FRH interaction being driven by charged/electrostatic interactions are not supported by the data. (1) Mutating the charged residues in vivo did not disrupt FRQ:FRH interaction, and (2) an enrichment of charge in the LOCATE hits and/or identification of the existence of charged blocks is not the same as experimentally demonstrating that the charge of residues within these newly identified interfaces is important for the FRQ:FRH interaction. Authors agree that the additional validation of the importance of these charged surfaces is justified but will be done in future studies. I see no issue with authors putting forward their best interpretation of their data (Figure 6e) in the Discussion, especially since they make it clear that other interpretations of their data are possible as well. However, as a result of the above limitations of their experiments, more careful language should be used in the abstract, introduction, and results sections. The authors should not both state that these validations will be performed in the future, and also claim them as discoveries in this manuscript.

-Thank you for clarifying your concerns, as it has allowed us to respond more appropriately. While the existence of charge blocks on FRQ and the effect of charge block ablation on clock function both support a role for these charge blocks, most likely through the regulation of the interaction of FRQ and FRH, we agree that this hypothesis is not directly proven in our work. In response, we have removed all remaining interpretation from the abstract, introduction, and results sections and have also modified the title of our manuscript to better reflect the main contribution of our paper. To this end, we have changed, removed, or added the following lines:

-Abstract:

- 1) Removed mention of multi-valent interactions (previous abstract line 34).
- 2) We also removed previous abstract lines 36-39, and lines 41, where we had mentioned the fuzzy-like complex model.

-Introduction:

- 1) We modified previous lines 114-116 from "LOCATE-based identification of binding sites within FRQ enabled direct insight into the molecular basis of the FRH/FRQ interaction..." to now read on lines 113-114 "LOCATE-based identification of *FRH* binding sites within FRQ *provided* insight into the FRH/FRQ interaction...".
- 2) Removed sentence about how our results are consistent with our proposed model of a multi-valent fuzzy-like complex (previous lines 108-111).

-Results:

- 1) Heading on Line 118: "LOCATE recapitulates a known FRQ SLiM and identifies novel interaction sites" now reads "LOCATE recapitulates a known FRQ SLiM"
- 2) Added text within lines 133-136 at the end of the first results section, highlighting the recapitulation of the FFD region.
- 3) Line 140-141 the phrase "...suggesting the FRQ/FRH interaction was more complex and involved regions beyond the FFD" was removed.
- 4) Heading on previous lines 144-145 changed from "FRH interacts predominantly with positively-charged FRQ peptides, uncovering significant clusters of positive residues within FRQ" to just be (current line 138) "FRH interacts predominantly with positively-charge FRQ peptides in the LOCATE assay", and we instead added a later heading relating to the significant clusters of positive charge "Clusters of positive residues are commonly occurring features in Negative-Arm Clock Proteins" (lines 159-160).
- 5) Lines 146-148 was adjusted from "To further characterize the overall binding behavior in our LOCATE assay and determine the underlying factors that dictate the interaction between FRQ and FRH, we focused on the top 10% of FRH-binding FRQ peptides." to now read on lines 143-145 "To further characterize the overall binding behavior *between FRQ peptides and FRH* in our LOCATE assay, we *investigated the characteristics of* the top 10% of FRH-binding FRQ peptides."
- 6) Previous line 275, removed end of sentence that could be seen as speculation, specifically removing "...the same interactions we expect to underlie the loss of interaction with FRH."
- 7) Line 299 modified from "...the FFD SLiM is distinct from other FRH-binding regions..." to be on line 277 "...the FFD SLiM is distinct from other FRH-binding *peptides*..."

-In addition, to clarify language related to the peptide array, we have modified the text to eliminate the terminology of "driven" electrostatic interactions, so these are also not potentially interpreted as relating to FRQ/FRH complex formation (which depends on the FFD region instead):

- 1) Line 277, removed "...tunes the *electrostatically-driven* intramolecular interaction of this peptide", and the full sentence now reads on lines 253-255 "Taken together, our simulations suggest that the loss of these three lysines *does not* significantly alter the accessibility of the nearby FCD-1 motif *and that the ablation of the charge in this region underlies the noted clock effects.*"
- 2) Lines 284-285 "While most FRH interaction with FRQ peptides in the LOCATE assay was *driven by* electrostatics..." changed to "While most FRH interactions with FRQ peptides in the LOCATE assay were *correlated with regions of positive charge, consistent with a role for electrostatics in these interactions ...*" (current lines 261-263).
- 3) Line 296-297 "A comparable analysis of the *electrostatically-driven* interaction near the FCD-1 region..." changed to lines 274-275 "A comparable analysis of the interaction near the FCD-1 region..."
- 4) Line 482 "Molecular interactions *driven by* intrinsically disordered regions..." is now "Molecular interactions *mediated by* intrinsically disordered regions..." (line 457)
- 5) Fig. 1 caption no longer includes a statement about the potential multivalent interaction between FRQ and FRH, but now simply states "LOCATE recapitulates a known FRQ SLiM".
- 6) The title of the Fig. 3 caption has also been changed to avoid using "electrostatically-driven" language, and now reads "There are significant charge clusters within FRQ and mutating a positive charge block affected clock period and output."

Statements like "clusters of positively charged residues within FRQ contribute to non-specific binding between FRQ and FRH," "While most FRH interaction with FRQ peptides in the LOCATE assay was driven by electrostatics," "Our method also revealed FRH can bind clusters of positively charged residues found within FRQ" or "FRH can bind clusters of positively charged residues found within FRQ" are not supported by the provided experimental data.

-For the quoted statements above, we have added, removed, or modified these statements as follows:

- 1) Original lines 33-34 quoted above "Further, we demonstrated that **clusters of positively-charged residues within FRQ contribute to non-specific binding between FRQ and FRH**" has been removed from the Summary.
- 2) Lines 284-285 "While most FRH interaction with FRQ peptides in the LOCATE assay was driven by electrostatics," was changed as outlined above to avoid the use of the term "driven".
- 3) It appears that the last two quotes above are from lines 104-105 "Our method also revealed **FRH can bind clusters of positively charged residues found within FRQ**" which has now been modified to read on lines 106-108 "Our *LOCATE* method also revealed FRH *binds to FRQ-based peptides that are positively-charged and that these positively-charged residues are significantly clustered within the FRQ sequence.*"

The authors themselves point out in the Results section that it is possible that mutations of lysines to alanines could modify the conformational ensemble of FRQ in a way that disrupts molecular interactions rather than being charge-driven, therefore their conclusions about the role of charge in the text need to be more carefully stated.

-We appreciate the reviewer's concern. To clarify our position, our conclusion regarding the role of charge was based not only on the overall charge of the original peptide containing the triple lysines but is also based on the scrambled peptides in our LOCATE assay maintaining binding with FRH (as opposed to the case of scrambled FFD-region peptides; Ext. Data Fig. 4e versus 4f). It is difficult for us to reconcile an

interpretation whereby KKK/AAA mutation disrupts the conformational ensemble to prevent binding, yet a complete scramble somehow does not disrupt the conformational ensemble. To further investigate this question directly, we additionally performed all-atom simulations, which again suggest that the KKK/AAA mutation does not lead to conformational changes that would affect the availability of neighboring regions (see Ext. Data Fig. 4a). This is also supported by the continued binding of CK1A to FRQ in the FRQ^{KKK/AAA} strain, with the FCD-1 region (right next to these mutated lysines) being required for CK1A binding². However, in response to the reviewer's concern, we have modified lines 243-245 and added to lines 251-255 to better reflect the contribution of the all-atom simulations and the corroborating evidence from the scrambles.

For the same reason, statements about this interaction being driven by a Fuzzy complex in the abstract and last paragraph of the introduction are problematic. It is the authors' hypothesis in light of inconsistencies *in vitro* and *in vivo*, but it is not an experimentally supported conclusion.

-We understand the reviewers desire to make clear the difference between our hypothesis and our data. In order to better highlight this, we have modified the abstract and introduction as outlined above. However, as we have discussed before, we don't believe our *in vitro* and *in vivo* results are necessarily inconsistent. In fact, our data that binding in the positively charged regions is not the principal driver of FRQ/FRH interaction is the insight that informed our multivalent model. To better highlight our hypothesis, we have included lines 401-403 about how microarrays can "capture a wide range of interaction affinities", and how our proposed model that incorporates both the *in vitro* and *in vivo* findings, lines 412-415, "The LOCATE data, in combination with a lack of change in the interaction between FRQ and FRH when a positive charge block was mutated, is consistent with positive charge blocks mediating multivalent FRQ/FRH interactions at secondary sites."

Previous concern #3:

Portions of the Results section are still motivated by hypothetical conclusions. The most problematic is chapter #2 (FRH interacts predominantly with positively charged FRQ peptides, uncovering significant clusters of positive residues within FRQ), which is motivated by the assumption that the charge is the driving force behind the FRQ:FRH interaction in LOCATE because it is enriched in the peptides that were identified as hits, although not validated. The most obvious example (but not the only one), is in the last paragraph of this section, where the authors move ahead with their interpretation despite the lack of experimental support: "Given the hypothesized charge clusters and blocks, we next wondered if these could be dynamically modulated." Because the authors never experimentally establish their molecular interpretation, the text is not scientifically sound. Authors should revise this section carefully and find a more accurate framing of their experimental results. I understand that some of these issues arose while authors tried to address reviewers concerns, however, it is authors' responsibility to work with the limitations of their data while ensuring readability and scientific soundness of the manuscript.

-Concern #3 is intricately intertwined with concerns #1 and #2, which we have addressed above. To demonstrate specific examples of the changes we have made to address concern #3, we would like to draw the reviewer's attention to the following sections:

- 1) Lines 147-149 brings our findings from Fig. 3a forward and discusses the binding behavior of FRH to the peptide microarray relative to the solved crystal structure of FRH. Further, on line 151, we bring up that the blue dots (positive NCP) within FRQ sequence being grouped is congruent with charge-patterning being a sequence feature for disordered proteins. In essence, we motivate

our examination of the charge blocks from data other than our hypothesis that these blocks are part of the FRQ/FRH interaction.

2) We then modified the text to separately look at our bioinformatic analysis, showing the sequence feature of positive charge clusters was significant within FRQ and PER orthologues. We also removed the sentence on lines 162-164 "Given the importance of positively charged peptides in our LOCATE assay, we speculated that an important feature in FRQ may be clusters of positively charged residues."

3) We also felt that the section following lines 213-214, "Given the hypothesized charge clusters and blocks, we next wondered if these could be dynamically modulated.", could also be considered as driven by hypothetical conclusions. To eliminate this concern, we have summarized the entire previous paragraph from 213-227 to now be one sentence on lines 200-203.

Previous concern #4:

My concern is not related to the presence of the final model in the discussion. My concern is related to the overall message of the abstract. Authors emphasize many IDR-related conclusions that are weakly supported and hypothetical, such as the biochemical basis of interactions, which are inconsistent in vitro and in vivo, rather than highlighting the novel unexpected roles in the molecular clock. Since authors argue in their previous response to reviewer #2 that the central novelty of their manuscript is the unexpected role of the described SLiMs in the molecular clock, the abstract should reflect that. As a result, this concern has not been addressed.

-Thank you for the clarification as to what section of the manuscript you were referring to. In concordance with the changes outlined above, we have changed our manuscript title, removed portions of previous lines 33-34, 36-39, and 41-42 in the abstract, which mentioned multi-valency or fuzzy complexes, and added updated lines 32-35, and 42-45 to allow the abstract to better highlight our novel findings related to the expanded FRH SLiM within FRQ, the separate roles that FRQ and FRH play in the clock, as well as the potential role that charge blocks play in the regulation of the molecular clock. We leave to the discussion the model related to how these charge blocks could work to control circadian regulation. Please note that we have also decided to move away from all "fuzzy"-related language in our discussion, as that terminology is not necessary for our proposed dynamic multi-valent model and is a term that continues to change in its definition within the disordered protein field (lines 427 and 441, and related Figure 6e caption).

Previous concern #5:

This is stylistic, and might be a minor concern. Consistent with my previous review, large parts of the manuscript devoted to IDR-related analyses in the Results section remain very long, but lack in impact. The most problematic is the Results section corresponding to Figure 3a-c, which takes up more than 2 pages of text (lines 153-212). This reduces overall readability. A more concise and focused message in this section in particular may increase the overall impact on the molecular clock field.

-We agree with the reviewer that this section is very detailed (lines starting at 153 in this version). This was purposeful, as during the review process it became clear to us, in part based on the good advice of this reviewer, that it was important to be thorough in our explanation of the elements related to disordered proteins so that the readers from the clocks field could more easily understand the importance

of the topic. However, we agree that this section can drag, and we don't want our readers to get lost in the forest for the trees. Therefore, to make this section more concise, we extensively edited the text. We also decided to summarize the predicted changes to charge blocks given previously determined FRQ phosphorylation sites and how this may affect predicted SLiMs for downstream interactors (as mentioned above, in response to concern #3). To this end, previous lines 213-227 have been condensed to lines 200-203 "Interestingly, the phosphosites that occur on FRQ over the circadian day have a greater potential to alter negative charge blocks rather than positive charge blocks along the length of FRQ, and overlapped with many predicted downstream SLiMs (Extended Data Figs. 2e-f)". In addition, we have removed some of the speculative language from this section in concordance with the above reviewer suggestions, which has also decreased the length of this portion of the results. (E.g. removed previous lines 153-154, 162-164, 171-174 shortened, removed 195-198, 200-205, 208-209). Because of the above changes, we feel this part of the results now relates a more focused and impactful message.

In total, we hope you agree that we have extensively edited the text to streamline the report and clarified what data we have gathered from our study versus what are our hypotheses based on these conclusions.

Reviewer #2 (Remarks to the Author):

The revised manuscript represents a notable improvement over the previous submission. The authors have addressed all the previously raised concerns, which enhanced scientific soundness and clarity of the paper. I applaud the authors' efforts and have no additional concerns.